# Single-trial modeling separates multiple overlapping prediction errors during reward processing in human EEG

Colin W. Hoy [1✉], Sheila C. Steiner[1] & Robert T. Knight[1,2]

Learning signals during reinforcement learning and cognitive control rely on valenced reward prediction errors (RPEs) and non-valenced salience prediction errors (PEs) driven by surprise magnitude. A core debate in reward learning focuses on whether valenced and non-valenced PEs can be isolated in the human electroencephalogram (EEG). We combine behavioral modeling and single-trial EEG regression to disentangle sequential PEs in an interval timing task dissociating outcome valence, magnitude, and probability. Multiple regression across temporal, spatial, and frequency dimensions characterized a spatio-tempo-spectral cascade from early valenced RPE value to non-valenced RPE magnitude, followed by outcome probability indexed by a late frontal positivity. Separating negative and positive outcomes revealed the valenced RPE value effect is an artifact of overlap between two non-valenced RPE magnitude responses: frontal theta feedback-related negativity on losses and posterior delta reward positivity on wins. These results reconcile longstanding debates on the sequence of components representing reward and salience PEs in the human EEG.

[1] Helen Wills Neuroscience Institute, University of California Berkeley, Berkeley, CA, USA. [2] Department of Psychology, University of California Berkeley, Berkeley, CA, USA. ✉email: hoycw@berkeley.edu

Adaptive behavior requires predicting relationships between stimuli, actions, and outcomes to decide which choices maximize reward. Predictive coding is a general computational framework that learns these mappings based on surprise as measured by prediction errors, computed as the difference between observed and expected outcomes. Reward prediction errors (RPEs) convey the valence (better or worse?) and magnitude (how surprising?) of a reward relative to expectations. RPEs are encoded by transient firing rate modulations of midbrain dopamine (DA) neurons[1–3], and early work in reinforcement learning (RL) established that simple and efficient model-free algorithms such as temporal difference learning can account for basic reward learning phenomena by using RPEs to update the expected value associated with a stimulus without the need for additional modeling of the influence of actions[4].

In contrast, sensorimotor and abstract cognitive control loops depend on non-valenced PEs to learn action-outcome contingencies[5]. For example, the predicted response-outcome (PRO) model asserts that the medial prefrontal cortex (MPFC) controls action selection by tracking salient, unexpected outcomes independent of valence[6]. Valenced and non-valenced PEs have complimentary contributions to learning[7], and current model-based reinforcement learning algorithms combine both valenced reward and non-valenced state, action, and outcome predictions to account for more complex behaviors by modeling relationships between an agent's actions and the environment[8–11]. Recent animal investigations of reward and control emphasize the multitude of learning signals represented by subpopulations of midbrain DA neurons[12], including aversive outcomes and stimulus salience independent of valence[13–15], as well as sensory, cognitive, and reward variables[16].

Non-invasive human electroencephalography (EEG) findings have identified a variety of learning-related event-related potentials (ERPs) and time-frequency signatures[17], but their specific relationships to reward and control PEs are still debated. Seminal early studies identified a posterior scalp positivity generated ~300 ms after detection of an infrequent stimulus called the P3[18,19] and a fronto-central negativity elicited ~80 ms after incorrect compared to correct responses called the error-related negativity (ERN)[20,21]. Subsequent extensive literature has revealed these ERPs to be part of large families of similar components. P3-style ERPs are characterized by slow ramping dynamics peaking from ~300–600 ms and in delta frequencies (~1–4 Hz)[22] with at least two different scalp topographies. The P3b has a posterior maximal topography and is elicited by detected events conveying various forms of salient information leading to working memory and model updates, while the P3a has an earlier latency, fronto-central topography and is generated by attention and orienting to novel, task-irrelevant stimuli[23–25] (see ref. [26] for review). Note that both the P3a and P3b are not unitary physiological events but rather reflect the summed activity of multiple intracranial sources[27]. ERNs are related to a family of faster latency (~200–300 ms) N2 negativities over fronto-central sensors generated in part by phase-locking in theta frequency (~4–8 Hz) MPFC activity triggered by unexpected events requiring behavioral adjustment[28–30] (for reviews, see ref. [31] for stimulus-locked N2s and ref. [32] for response-locked ERNs).

Reward feedback conveys multiple informative variables with varying salience and elicits both N2- and P3-like ERPs that overlap in time and space, leading to longstanding debates over which components track different aspects of feedback. Foundational studies focused on the N2-like feedback-related negativity (FRN) occurring ~200 ms at frontal sites after loss feedback compared to wins[33,34]. An early, influential RL theory (originally called the RL-ERN theory) proposed the FRN represents valenced, quantitative RPE value driven by midbrain DA

projections to MPFC[35]. This hypothesis predicts FRN sensitivity to all the feedback properties that determine RPEs: outcome valence, magnitude, and probability[36]. However, two recent meta-analyses found mixed evidence for magnitude and probability effects[37,38]. Reports of larger FRNs to unexpected positive outcomes[29,39–42] led to an alternative account called the salience theory, which proposes the FRN represents the degree of surprise of an outcome regardless of valence, similar to non-valenced action-outcome PEs driving cognitive control in the PRO model.

A third prominent proposal called the independent coding hypothesis posits the FRN represents binary reward valence instead of scalar RPE value, while the subsequent P3 tracks non-valenced RPE magnitude[43–45]. This interpretation is complicated by more recent observations of a P3-like positivity called the Reward Positivity (RewP) that tracks RPE magnitude specifically on positive outcomes (see ref. [46] for review)[47–52] that overlaps in space and delta frequencies with other non-valenced P3 components[53,54]. Importantly, this suggests losses and wins generate distinct FRN and RewP ERPs with opposite polarities that interact to some degree depending on their overlap in time and space[48,55]. As a result, it remains unclear after decades of research whether FRN and/or RewP ERPs are driven by valenced RPEs, non-valenced salience PEs, or one of the valenced/non-valenced input variables contributing to these PEs (e.g., outcome valence, magnitude, or probability).

An important challenge in resolving this debate is disentangling overlapping ERP components. For example, the FRN is commonly measured by averaging ERP amplitude across time, but the epochs used in mean window analyses cover both classic N2 and P3 windows[38]. The FRN is also often measured by the peak-to-peak amplitude difference between the N2 and the preceding P2 positivity to account for influences of early P3 ramping. However, individual ERP peaks are variable and may not correspond to unique neural sources[56–58], rendering their use as reference measures questionable. For example, the P2 shows confounding effects of surprising positive reward[54,59]. Difference waves are commonly used to isolate target variables such as valence by subtracting ERPs across conditions matched for confounding variables (e.g., magnitude or probability). Indeed, win-loss difference waves are commonly used as the operational definition of the FRN and RewP[38,54,60,61]. However, this subtraction logic can determine neither which ERP nor which condition was modulated and thus is not well suited to unraveling the dual multiplicity of ERPs and learning signals, particularly since the FRN and RewP may have distinct neural mechanisms supporting different computational roles[52,62,63]. Here, we use the term FRN to refer to the early, feedback-locked frontal N2-like ERP that is most prominent on but not specific to losses and the term RewP to refer to the subsequent feedback-locked P3-like ERP specifically following wins. We return to the definitions of FRN and RewP components and their relationship to other ERPs and difference wave contrasts in the discussion.

The overlap of ERPs in time-domain analyses has made time-frequency decompositions an important tool for separating the FRN and RewP into theta and delta frequencies, respectively. Several studies have shown theta is sensitive to negative RPEs[51,53,63,64], but it has also been reported to track non-valenced probability and magnitude[65–67]. Likewise, delta activity is linked to both non-valenced surprise[24,25,68] and positive RPEs[46,51,53,63,69]. These mixed results highlight how individual measures of neural activity may be insufficient to distinguish between ERP components such as the RewP, P3a, and P3b that overlap in one or more dimensions (e.g., frequency) but correspond to distinct cognitive variables.

Single-trial modeling methods have provided clarity into the theoretical debates in reward processing EEG signals. RL models

estimate latent cognitive variables such as reward expectations, which can change the subjective meaning of and ERP response to reward feedback[36,70,71]. Importantly, model-based single-trial correlation or regression analyses provide enhanced statistical power compared to traditional categorical statistics applied to condition-averaged data[72,73], which can be leveraged to map the evolution of cognitive variables in high resolution across multiple timepoints, channels, and frequencies. This framework has been used to separate overlapping components[51,63,66,69,74]. In particular, this approach enables data-driven discovery of RL variable representations in EEG data that are not time-locked to ERP peaks[75,76], and allows formal model comparisons between competing hypotheses[24,25,68]. Here we use these methods to compare the predictions of the main competing hypotheses across the different measurements of the FRN and RewP.

Our goal was to combine these modeling and signal analysis tools to provide a comprehensive assessment of the core theoretical and measurement issues underlying the FRN and RewP debate. We start by estimating PEs from individual participant behavior in an interval timing task designed to dissociate the valence, magnitude, and probability of outcomes. We then use formal model comparisons to test the predictive power of outcome and PE features central to RPE, salience, and independent coding hypotheses using mixed-effects multiple regression analyses applied across temporal, spatial, and spectral dimensions of wins and losses to separate overlapping components in initial and replication cohorts. To relate our results to previous EEG literature, we perform analogous modeling of mean window and peak-to-peak FRN and RewP metrics, in addition to quantifying the overlap in our reward feedback ERPs based on correlations with canonical N2 and P3 benchmark ERPs measured in a three-tone oddball task collected from a subset of the same participants.

We found that when modeling wins and losses together as done in standard analyses, early, frontal theta activity underlying the FRN is best described by valenced RPE value, while non-valenced RPE magnitude and probability effects drive later delta activity consistent with P3 ERPs. Model comparisons show these data are better explained by RL-based PEs than outcome features and confirm the importance of predictive coding principles. Mean window and peak-to-peak FRN analyses replicated these RPE value effects but also showed non-valenced effects and were unstable across the two cohorts. These results suggest the FRN represents a scalar, quantitative RPE while two P3 ERPs encode non-valenced RPE magnitude and outcome probability. However, these conventional analyses combining wins and losses confound the FRN on negative trials and the RewP on positive trials, and comparisons to oddball task ERPs suggest a mixture of N2 and P3 contributions to the RPE value window. Indeed, modeling EEG amplitude, topographies, and time-frequency power after separating wins and losses reveals that the valenced RPE value effect is an artifact of non-valenced RPE magnitude driving two overlapping FRN and RewP components in theta for losses and delta for wins, respectively, while the late frontal probability effect in delta is stable across outcomes. Finally, we use subjective ratings obtained in a follow-up behavioral experiment to confirm these EEG results cannot be explained by subjective biases in reward contingencies. Collectively, these results provide strong evidence that human EEG following reward feedback is composed of a sequence of multiple overlapping neurophysiological signatures best accounted for by specific PEs in a predictive coding framework.

## Results

We collected and analyzed EEG from 32 cognitively normal young adults split into initial ($n = 15$) and replication ($n = 17$) cohorts.

Participants performed an interval timing task designed to dissociate the key variables underlying the PEs central to the debate between RL, salience, and independent coding accounts: outcome valence, magnitude, and probability. At the beginning of each trial, participants saw a target zone cue whose size indicated the temporal range of responses tolerated as correct (Fig. 1a). Participants then estimated the temporal interval by means of extrapolation from visual motion, and received audiovisual feedback indicating their reaction time (RT) and whether it was within or outside the tolerance (i.e., a win or loss). After each trial, the error tolerance was titrated by two staircase algorithms (Fig. 1b) to clamp accuracy at $82.7 \pm 1.7\%$ and $18.1 \pm 2.5\%$ (mean ± SD) in easy and hard blocks, respectively (Fig. 1c). This design dissociates outcome valence and probability to separate valenced and non-valenced PE features by comparing surprising wins and losses. Neutral outcomes with no RT feedback were also delivered on a random subset of 12% of trials to manipulate outcome magnitude as another source of surprise.

**Behavioral modeling**. To directly compare the predictive power of RL, salience, and independent coding theories, we used computational modeling of individual participant behavior to derive single-trial estimates of valenced RPE value, as well as two sources of salience: non-valenced RPE magnitude and outcome probability. For each participant, we used logistic regression to fit the relationship between the interval tolerance and binary win/loss outcomes across the entire session (Fig. 1d; see inset for group model fits). The resulting model yields the probability of that participant winning for any given tolerance, which was then linearly scaled to the range of rewards (1, 0, and −1 for winning, neutral, and losing outcomes) to quantify expected value for every trial. We then contrasted expected value with actual outcomes to obtain single-trial RPE values and derived the absolute value of RPEs to obtain RPE magnitudes. Outcome probability was determined by the frequency of each outcome in each condition. Notably, RPE values for neutral outcomes were non-zero and switched valence across blocks (negative for easy and positive for hard blocks; see model predictions in Supplementary Fig. 1a), suggesting they could be interpreted as omissions of the expected outcome.

To compare this RL model to simple win/loss contrasts standard in the FRN and RewP literature that do not account for predictive coding, we also computed an outcome-based model composed of reward value (1, 0, and −1), reward magnitude (absolute value of reward value), and outcome probability. Finally, to test the hypothesis that the FRN tracks binary valence but not scalar value or magnitude, we added a modified outcome model that replaced the outcome value on neutral trials with valence based on reward omission relative to expected value (see Supplementary Fig. 1b for outcome-based model predictions).

**Single-trial regression reveals a spatio-temporal cascade of PE components**. According to the RL theory, the valenced RPE values derived from our behavioral model should predict FRN amplitude[35,36], while the independent coding hypothesis suggests the FRN is sensitive only to binary reward valence and not scalar value (i.e., combined valence and magnitude)[43–45]. In contrast, the salience theory predicts that FRN amplitude should scale with how surprising each outcome is[6,39,66], which increases with non-valenced RPE magnitude and decreases with outcome probability. Importantly, disentangling these outcome and PE features and resolving this debate depends on addressing the overlapping ERP component problem that confounds traditional mean window and peak-to-peak amplitude measurements[60]. Here, we leverage known timing differences between early FRN and later P3 activity

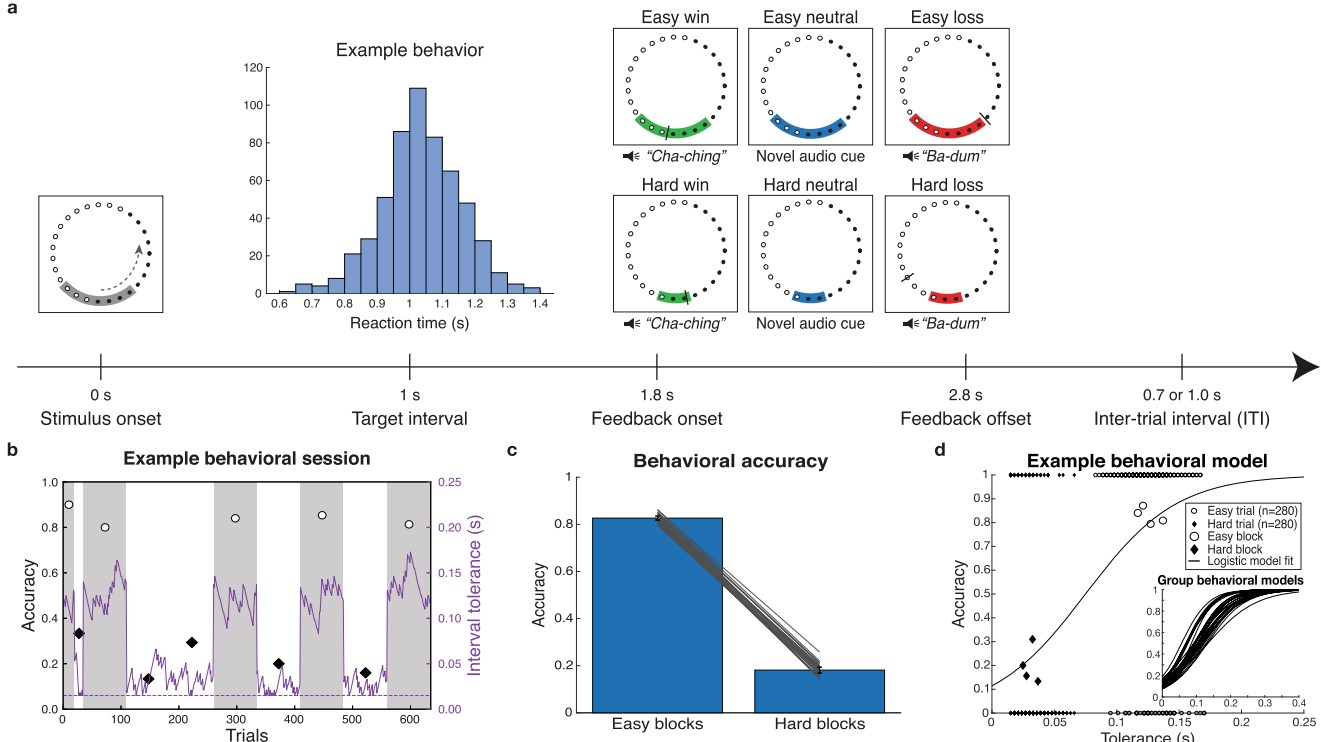

**Fig. 1 Task design, performance, and behavioral modeling of prediction errors. a** Participants pressed a button timed to the estimated completion of lights moving around a circle. The gray target zone cue displayed error tolerance around the 1 s target interval. An example participant RT distribution is centered at the target interval. Audiovisual feedback is indicated by the tolerance cue turning green for wins and red for losses. A black tick mark displayed RT feedback. On 12% of randomly selected trials, blue neutral feedback was given with no RT marker. **b** Example recording session for one participant for training (first 35 trials) and experimental blocks. Staircase adjustments of tolerance are plotted as a solid purple line, and the dotted purple line indicates the minimum bound on tolerance at ±15 ms. Accuracy for easy and hard blocks is plotted as white circles on gray backgrounds and black diamonds on white backgrounds, respectively. **c** Separate staircase procedures resulted in group accuracies of (mean ± SD) 82.7 ± 1.7% for easy and 18.1 ± 2.5% for hard blocks. Error bars indicate standard deviation across participants, with individual participant ($n = 32$ participants) accuracy overlaid as gray lines. **d** Tolerance and outcome data for the same example participant. Larger markers show block-level accuracy; smaller markers show binary single-trial outcomes. Model fit using logistic regression provides single-trial estimates of win probability, which can be converted to expected value. Inset shows win probability curves across all 32 participants.

by predicting single-trial evoked amplitude at every time point from 50 to 500 ms post-feedback using mixed-effects multiple regression analyses to adjudicate between different PEs. Within this framework, we use formal model comparisons to test whether RL models combining expected value, RPE value, RPE magnitude, and outcome probability predict ERP amplitudes better than standard models composed of outcome magnitude, probability, and either scalar value or binary valence. These analyses were conducted separately at frontal Fz and posterior Pz electrodes to assess the FRN and longer latency positivities.

Grand-average ERPs show FRN peaks ~200–250 ms post-feedback at frontal electrode Fz and P3 peaks ~300–350 ms at posterior Pz (Fig. 2a,b; see also Supplementary Fig. 9a). Model coefficients are plotted below in Fig. 2c,d for the best performing model, which includes RL features of expected value, RPE value, RPE magnitude, and probability (see Supplementary Fig. 2 for difference wave contrasts for each of these variables). The most prominent result is a large effect of RPE value peaking in the FRN window at 216 ms in electrode Fz ($\beta_{max} = 4.572$, $q_{FDR} < 10^{-10}$; Fig. 2c). In accordance with the RL theory, this positive model coefficient indicates that more negative RPE values are associated with more negative amplitudes. In other words, larger FRNs with more negative amplitude are associated with worse-than-expected outcomes, and better-than-expected outcomes drive more positive amplitude. The RPE value effect decreases as the FRN subsides, and a significant positive RPE magnitude effect emerges.

This RPE magnitude effect is maximal in electrode Pz at 308 ms ($\beta_{max} = 1.703$, $q_{FDR} < 10^{-10}$; Fig. 2d), indicating larger non-valenced RPE magnitudes are associated with larger P3 amplitudes. This result replicates the non-valenced effect of magnitude matching a posterior P3b predicted by the independent coding hypothesis[43], but this analysis included both positive and negative outcomes and cannot disambiguate potential contributions of a RewP specific to positive RPEs.

The temporal coincidence of significant model coefficients for RPE value and magnitude in the epoch between FRN and P3 peaks suggests that previous findings supporting the salience theory could be explained by component overlap confounds, particularly since FRN and RewP amplitude is commonly quantified as the mean amplitude from ~228–344 ms[38], an epoch that encompasses both FRN and P3 activity. To compare our time-resolved single-trial regression analyses to metrics more commonly used in the field, we computed mean window and peak-to-peak estimates of the FRN, as well as mean window measures of the P3 (Supplementary Fig. 3a). As predicted, RL model coefficient results using traditional mean window and peak-to-peak estimates of the FRN confirmed the strong RPE value effect. However, they also show significant but divergent non-valenced effects, with mean window predicted by probability and peak-to-peak predicted by RPE magnitude (Supplementary Fig. 3b,3c). Importantly, these conflicting non-valenced FRN effects using traditional methods were unreliable across

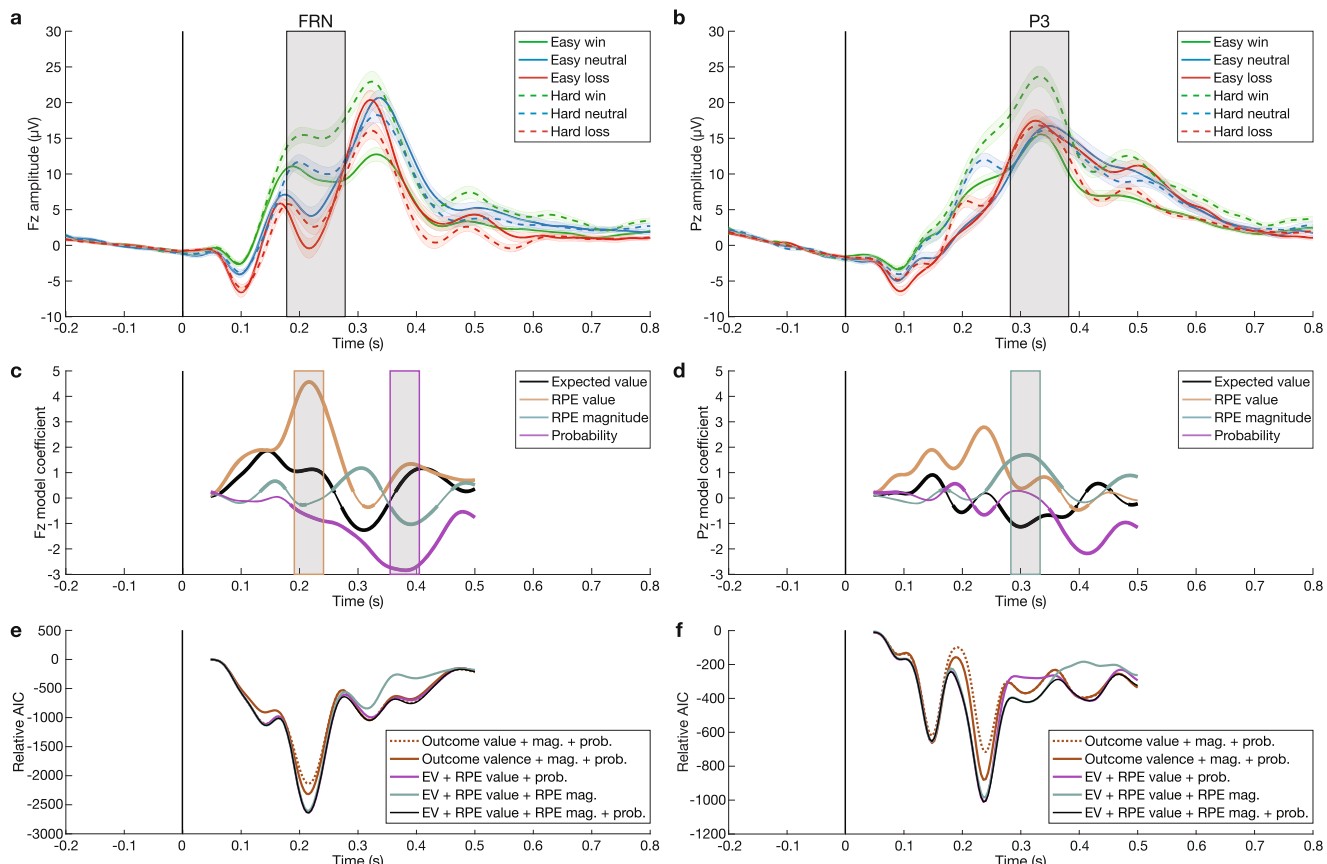

**Fig. 2 Single-trial modeling of ERP amplitude reveals a sequence of prediction errors. a** Feedback-locked grand-average ERPs at Fz plotted for each condition, with shaded error bars indicating standard error of the mean across 32 participants. FRN is evident as prominent negative deflections at Fz ~200 ms post-feedback. Vertical black line shows feedback onset. Gray shading shows 100 ms window used to average FRN amplitude. **b** Same for posterior electrode Pz, which shows large P3 positivities at ~300 ms. Gray shading shows 100 ms window used to average P3 amplitude. **c** Model coefficients from single-trial multiple regression at each time point from 50 to 500 ms at Fz show a strong early peak of valenced RPE in the FRN window, followed by later probability effect in the P3 time range. Bolding indicates significant timepoints ($q_{FDR} < 0.05$; $n = 32$ participants). Gray shading shows 50 ms windows used to average ERP amplitude at maximal RPE value and probability effects. **d** Same for electrode Pz. Note the increased non-valenced RPE magnitude coefficient in the P3 window. Gray shading shows 50 ms window used to average ERP amplitude at the maximal RPE magnitude effect. **e** Comparison of model performance at Fz over time via Akaike Information Criteria (AIC; more negative indicates higher performance) relative to baseline model. RL model with RPE value beats outcome-based models in the FRN window, and model performance drops in the P3 window when probability is excluded. **f** Same for Pz. RL model performance drops during P3 peak window when RPE magnitude is excluded, while RL model performance drops during late positivity window when probability is excluded. See Supplementary Fig. 5 for RL model performance ($R^2$) across electrodes, which peaks in the FRN time window for Fz and the P3 time window for Pz.

replication cohorts (see Supplementary Table 1 for RL model results across cohorts for FRN and P3 mean window and peak-to-peak metrics). Similarly, RPE magnitude was significant in the RL model regression for the P3 mean window analysis at Pz in both cohorts (Supplementary Fig. 3d), but RPE value was significant in only one cohort, confirming that mean window and peak-to-peak metrics are less reliable.

Finally, the probability predictor reveals a significant relationship to ERP amplitude that peaks later at 380 ms in Fz ($\beta_{max} = -2.842$, $q_{FDR} < 10^{-10}$; Fig. 2c). Observing this late, frontal positivity in response to unlikely outcomes was possible because we dissociated outcome probability and RPE magnitude as two distinct sources of salience.

Time-resolved model comparisons for Fz and Pz (Fig. 2e,f, respectively) are plotted as Akaike Information Criterion (AIC) values relative to a baseline model containing only random intercepts capturing each participant's mean amplitude across conditions. Lower AIC values mean better model performance. Figure 2e shows that RL-based models including RPE value capture more variance in EEG amplitude during the FRN window

at Fz than outcome value or valence models (see Supplementary Fig. 4c–f for outcome-based model coefficients; see Supplementary Table 2 for AIC model comparison values averaged within peak model coefficient windows). The model with binary outcome valence performs better than the outcome-based model with scalar value, and these model results hold for both mean window and peak-to-peak estimates of FRN amplitude (Supplementary Fig. 3e,f). The only difference between outcome value and outcome valence models is whether neutral trials in easy and hard blocks are treated as outcomes with identical values (zero) or as omissions of expected rewards with opposite valence (1 or −1). Similarly, Fig. 2f shows that the RL-based models outperform the outcome value and valence models at Pz throughout the FRN and P3 epochs, which is confirmed by model comparisons using the mean window estimates of P3 amplitude in Supplementary Fig. 3g. These results confirm that FRN and P3 ERPs are best viewed through the predictive coding lens of PEs.

Since RPE magnitude and probability are correlated (see Methods), we also used model comparisons as a control to examine whether the variance explained by these two non-

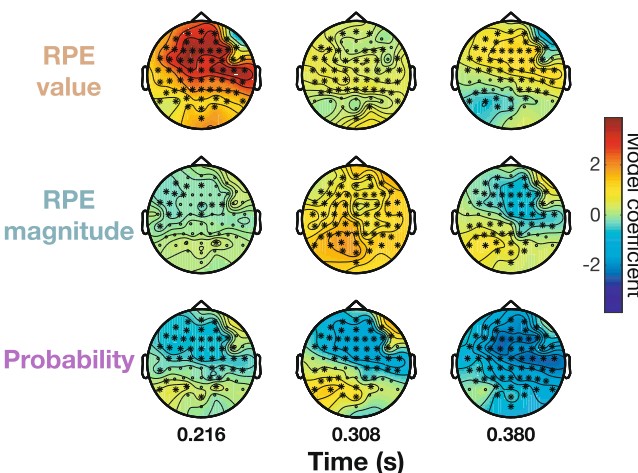

**Fig. 3 Spatio-temporal dynamics of prediction errors across ERP scalp topographies.** Single-trial regression over all 64 electrodes is computed for three 50 ms windows centered on the largest peak in model coefficients for RPE value (216 ms at Fz), RPE magnitude (308 ms at Pz), and probability (380 ms at Fz) from Fig. 2. Stars indicate significant electrodes ($q_{FDR} <$ 0.05; $n = 32$ participants). Valenced RPE value shows a frontal distribution in the early window (top left). Non-valenced RPE magnitude is maximal at posterior electrodes in the middle window (center). Probability is maximal at fronto-central sensors in the late window (bottom right). See Supplementary Fig. 6 for evoked potential voltage topographies.

valenced salience PEs dissociated in time and space. When the RPE magnitude predictor is excluded, RL model performance drops at Pz near the peak of the P3 (Fig. 2f). When the probability predictor is excluded, performance drops later during the downslope of the P3 at Pz and from ~350–450 ms at Fz (Fig. 2e,f; model coefficients plotted in Supplementary Fig. 4g–j), confirming that these two sources of salience correspond to separable EEG components (see also dissociations with Oddball ERP correlations below).

Collectively, these results characterize a cascade of multiple PEs unfolding during the FRN and RewP epochs in reward processing EEG, starting with an early, frontal, valenced RPE value signal in the FRN time window, followed by a later, posterior, non-valenced RPE magnitude effect in the P3 window, and finally a later, fronto-central probability effect. These findings replicate evidence supporting the RL account of the FRN as a scalar, quantitative RPE instead of binary valence[35–37] and the independent coding proposal's separation of early valenced effects in the FRN window from two later non-valenced P3 effects[43,45]. We reassess the interpretations of these effects after characterizing their spatial and frequency distributions, their correspondence with benchmark N2 and P3 ERPs from the oddball task, and most importantly when separating losses and wins to avoid confounding overlapping FRN and RewP ERPs.

**Scalp topography dynamics delineate PE components.** To further disentangle the spatial topographies of sequential PE effects and relate them to known N2 and P3 scalp distributions, we applied single-trial multiple regression analysis across all electrodes in three 50 ms windows centered on the peaks of each model coefficient from the time-resolved analysis (see highlighted windows in Fig. 2c,d). Model coefficients for valenced RPE value, non-valenced RPE magnitude, and outcome probability are plotted as scalp topographies in Fig. 3 (see Supplementary Fig. 6 for evoked amplitude topographies by condition). This analysis confirmed that the largest effect was the valenced RPE value in the early window at anterior frontal sites ($\beta_{max} = 3.887$ in the 216

ms window at electrode F1, $q_{FDR} < 10^{-10}$), which then dropped off in magnitude in the middle window before a smaller resurgence in the late window. The non-valenced RPE magnitude effect was maximal in the middle 308 ms window at posterior parietal electrodes ($\beta_{max} = 1.859$ at electrode PO3, $q_{FDR} < 10^{-10}$). Finally, the probability effect was focused in fronto-central electrodes in the later 380 ms window ($\beta_{max} = -2.523$ at electrode C4, $q_{FDR} < 10^{-10}$). The spatio-temporal distributions of these effects confirms the association between valenced RPE value and the early FRN epoch in frontal electrodes[34,35,74], while non-valenced RPE magnitude shows a posterior, parietal distribution matching the P3b[24,26].

**Single-trial regression of time-frequency power dissociates PE effects in theta and delta bands.** Although the N2/FRN and P3/RewP are defined as ERP phenomena, their waveform characteristics are associated with theta (4–8 Hz) and delta (1–4 Hz) frequencies, respectively[26,30,51,65,77]. To further dissociate contributions of RPE value, RPE magnitude, and probability to these overlapping components, we extracted feedback-locked time-frequency representations (TFRs) of evoked power at Fz (Supplementary Fig. 7) and Pz (Supplementary Fig. 8). Single-trial multiple regression with our RL model across wins and losses revealed a strong negative relationship between RPE value and theta power ($\beta_{max} = -1.405$ at [292 ms, 6 Hz] in electrode Fz, $q_{FDR} < 10^{-10}$; Fig. 4a). The delayed peak of this theta effect relative to the FRN latency highlights the spread of this RPE value effect spanning across several cycles of theta (see also theta frequency fluctuations of RPE value coefficients in Fig. 2c). In contrast, RPE magnitude significantly predicted posterior delta power ($\beta_{max} = -0.447$ at [260 ms, 3 Hz] in electrode Pz, $q_{FDR} = 5.05 \times 10^{-10}$; Fig. 4b), consistent with the upward ramp of the P3. Probability best predicts 4 Hz power at 392 ms post-feedback ($\beta_{max} = -0.871$ in electrode Fz, $q_{FDR} < 10^{-10}$; Fig. 4a). Overall, more negative RPEs predicted stronger theta power in accordance with the RL theory of the FRN[35], while delta power associated with P3 ERPs increased with larger RPE magnitudes and more unlikely events as predicted by the independent coding hypothesis[43].

**Correlations with Oddball task ERPs quantify contributions of overlapping components.** To further disambiguate the contributions of overlapping FRN and P3 ERPs to RPE value, RPE magnitude, and probability effects, we compared feedback-locked ERPs in the Target Time task to reference N2 and P3 ERPs in a canonical Oddball task. A subset of the Target Time participants ($n = 22$) performed a three-tone Oddball task in which they attended to a stream of audiovisual stimuli and were instructed to press a button after rare targets (12.25% of trials) among common standard (75.5% of trials) and rare, task-irrelevant novel stimuli (12.25% of trials; see Methods for details). Figure 5a shows frontal Oddball ERPs at Fz with a prominent N2c in the target condition and a smaller N2b in the novel condition, both associated with control allocation in response to unexpected rare stimuli but differing in task contingencies[31,78]. Posterior Oddball ERPs at Pz in Fig. 5b and spatial topographies in Fig. 5c show novel stimuli elicit a central P3a associated with bottom-up orienting of attention, and target stimuli elicit a classic posterior P3b related to top-down model updating[26]. Since the timing, topographies, frequency characteristics, and potential intracranial sources of the FRN and RewP are shared with ERPs in the larger families of N2 and P3 components, respectively[31,47,67,68,78–80], we used individual participant ERPs averaged in a 50 ms window centered on the grand-average N2 and P3 peaks in target and novel Oddball conditions as benchmarks to determine the relative contributions

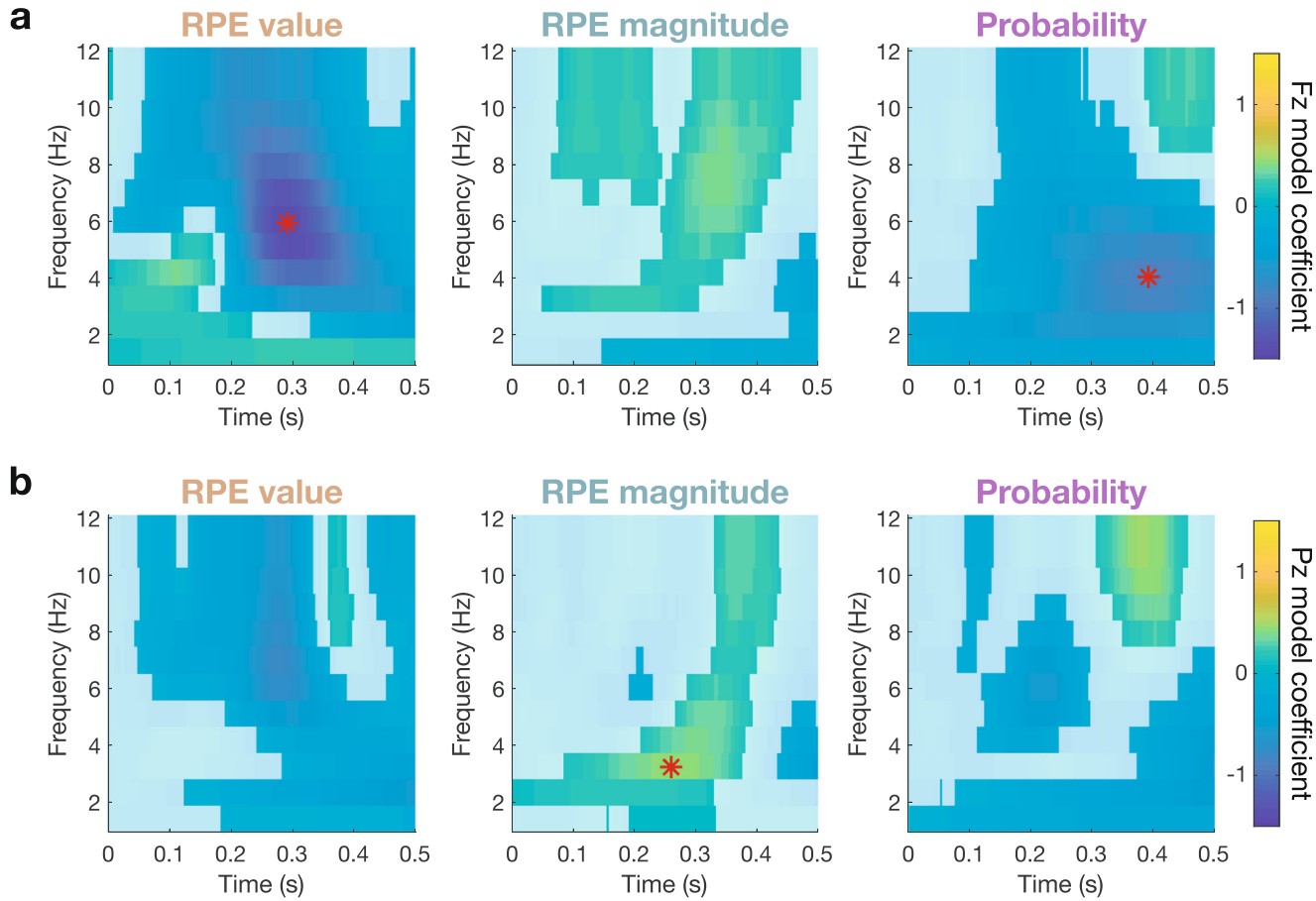

**Fig. 4 Time-frequency power signatures of prediction errors. a** Model coefficients fit to evoked power at each time-frequency point for frontal electrode Fz, with nonsignificant points ($q_{FDR} > 0.05$; $n = 32$ participants) plotted opaquely. Red stars indicate maximal coefficients for each model predictor across both electrodes. Valenced RPE value coefficients peak in frontal theta power, and non-valenced probability coefficients peak later in frontal delta power. **b** Same for Pz. Non-valenced RPE magnitude coefficients peak in posterior delta power.

of N2-like and P3-like activity to ERPs underlying the PE effects observed in our multiple regression analyses (as highlighted by colored window overlays in Fig. 2c,d).

Group-level correlations between Oddball ERPs and Target Time condition ERPs averaged at Fz in the RPE value peak window showed significant relationships between both Novel N2b and Target N2c ERPs and negative RPE conditions, with weaker but significant relationships between the stronger Target N2c and positive valence RPE conditions (Fig. 5d). These relationships suggest strong contributions of N2-like activity to the RPE value epoch in the Target Time task, particularly for conditions requiring control allocation such as button presses to targets in the Oddball task and adjustments in RTs following losses in the Target Time task (see elevated FRN mean window and peak-to-peak estimates in Supplementary Fig. 3a). However, significant correlations between the fronto-central P3a from the novel Oddball condition and four out of six Target Time conditions indicate additional influences of P3-like activity, reinforcing the risks of interpreting reward processing in the FRN/RewP epoch as a unitary phenomenon.

Comparisons between Oddball and Target Time ERPs at Pz in the RPE magnitude window show significant correlations specifically with Novelty P3a and Target P3b ERPs (Fig. 5e). Interestingly, the fronto-central Novelty P3a correlated with all Target Time conditions except hard losses, while the posterior Target P3b correlations were weaker and only significant for wins and easy neutral outcomes, despite these analyses being conducted at posterior Pz where the RPE magnitude effect was

maximal. No Oddball ERPs correlated significantly with Target Time ERP amplitudes in the later probability window at Fz (Fig. 5f). These results suggest the RPE magnitude effect is mainly driven by P3-like activity, while the later frontal probability effect has no clearly analogous ERP in the Oddball task.

**Separating outcomes by valence dissociates FRN and RewP RPE magnitude effects.** The mixture of N2 and P3 contributions to the RPE value window identified by Oddball ERP correlations revives concerns that the sequence of PE effects described above may be confounded by component overlap. In particular, our multiple regression results thus far account for competition between valenced and non-valenced PE predictors, but like difference waves, they still rely on contrasts between negative and positive outcomes that cannot distinguish between overlapping ERPs. To disentangle the roles of FRN and RewP ERPs in the sequence of PEs described above, we repeated our ERP and TFR multiple regression analyses separately for negative (easy loss, easy neutral, and hard loss) and positive (easy win, hard win, and hard neutral) outcomes using only RPE magnitude and probability predictors to avoid multicollinearity.

Model coefficients at Fz confirm that RPE magnitude exerts opposite effects across negative and positive valenced outcomes in the FRN window (Fig. 6a,b). On negative outcomes, the RPE magnitude predictor shows a significant negative effect in the FRN window only at frontal electrode Fz ($\beta_{max} = -1.833$ at 216 ms, $q_{FDR} < 10^{-10}$; Fig. 6a) but also shows notable rhythmic

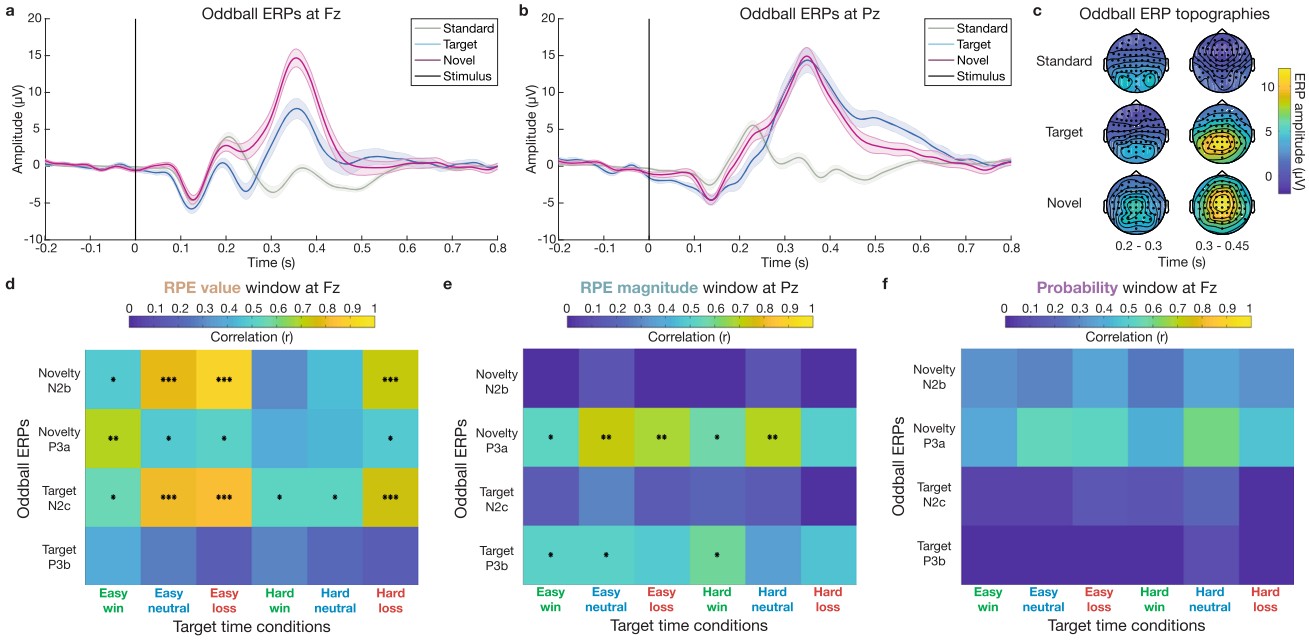

**Fig. 5 ERP comparison between Target Time feedback and 3-tone Oddball tasks. a** Stimulus-locked grand-average Oddball ERPs at Fz plotted for each condition, with shaded error bars indicating standard error of the mean across 22 participants. N2 is evident ~250 ms in target and to a lesser extent in novel conditions. **b** Same for Pz. P3 is evident at ~350 ms in target and novel conditions. **c** Spatial topographies of grand-average ERPs averaged during N2 (200–300 ms) and P3 (300-450 ms) windows. Note the posterior distribution of the Target P3b and central distribution of the Novel P3a. **d** Correlation matrix between N2 and P3 ERPs in Oddball task and mean amplitude of 50 ms window centered on maximal RPE value effect at Fz in Target Time conditions. Asterisks indicate significance at $q_{FDR} < 0.05$ (*), $q_{FDR} < 0.01$ (**), and $q_{FDR} < 0.001$ (***) across 22 participants. Note strong correlation between Oddball N2s and Target Time conditions with negative valence, especially for the Oddball Target N2c. **e** Same but correlating N2 and P3 amplitudes from Oddball task with mean amplitude of 50 ms window centered on maximal RPE magnitude effect at Pz in Target Time conditions. **f** Same but correlating N2 and P3 amplitudes from Oddball task with mean amplitude of 50 ms window centered on maximal probability effect at Fz in Target Time conditions.

fluctuations such that the maximal effect is at a second peak at 384 ms ($\beta_{max} = -1.953$ at Fz, $q_{FDR} < 10^{-10}$). On positive outcomes, RPE magnitude shows a sustained positive ramp that builds up to a maximum in the P3 window at posterior site Pz ($\beta_{max} = 2.743$ at 312 ms, $q_{FDR} < 10^{-10}$; Fig. 6b), with a similar but smaller significant effect at frontal Fz ($\beta_{max} = 2.065$ at 272 ms, $q_{FDR} < 10^{-10}$). In contrast, the late probability effect has a consistent sign and frontal distribution across positive and negative domains (positive: $\beta_{max} = -2.646$ at 336 ms in electrode Fz, $q_{FDR} < 10^{-10}$; negative: $\beta_{max} = -3.742$ at 392 ms in electrode Fz, $q_{FDR} < 10^{-10}$).

Modeling ERP amplitude topographies averaged in 50 ms windows centered on the peaks of the original RPE value, RPE magnitude, and probability effects in Fig. 2 confirms the distinct spatio-temporal dynamics of RPE magnitude effects for negative and positive feedback. For negative outcomes, RPE magnitude model coefficients show significant negative effects in fronto-central sensors in the early FRN window at 216 ms ($\beta_{max} = -1.556$ at electrode F2, $q_{FDR} < 10^{-10}$) which dissipate in the middle P3 window and return to their strongest levels at the late 380 ms window ($\beta_{max} = -1.684$ at electrode C4, $q_{FDR} < 10^{-10}$; Fig. 6c). For positive outcomes, RPE magnitude shows significant positive model coefficients that are maximal at central and posterior sites in the middle P3 window ($\beta_{max} = 2.683$ at electrode CP1, $q_{FDR} < 10^{-10}$; Fig. 6d). As in the ERP time-domain analyses, the probability predictor shows a significant negative effect strongest in the late window at fronto-central sensors regardless of valence (negative: $\beta_{max} = -3.093$ at 380 ms in electrode F1, $q_{FDR} < 10^{-10}$, see Fig. 6c; positive: $\beta_{max} = -2.099$ at 380 ms in electrode C4, $q_{FDR} < 10^{-10}$, see Fig. 6d).

These ERP results suggest qualitative differences in responses to positive and negative feedback, and separate regressions of TFR power for positive and negative outcomes reveal a dissociation of theta and delta power underlying the FRN and RewP components driving these RPE effects. Specifically, RPE magnitude shows a strong positive effect in frontal theta frequencies on negative trials ($\beta_{max} = 0.823$ at [292 ms, 7 Hz] in electrode Fz, $q_{FDR} < 10^{-10}$), but this effect shifts to posterior delta power on positive outcomes ($\beta_{max} = 0.871$ at [260 ms, 3 Hz] at Pz, $q_{FDR} < 10^{-10}$). Again, the late probability effect shows a consistent negative effect strongest in delta frequencies at Fz that is stable regardless of valence (negative outcome: $\beta_{max} = -0.734$ at [416 ms, 4 Hz] at Pz, $q_{FDR} < 10^{-10}$; positive outcomes: $\beta_{max} = -1.057$ at [388 ms, 4 Hz] at Pz, $q_{FDR} < 10^{-10}$). In sum, the large valenced RPE value effect seen when contrasting wins and losses in the FRN/RewP window is composed of the superposition of two separate non-valenced RPE magnitude effects: early frontal theta activity drives negative FRN amplitudes on negative outcomes, and prolonged, more posterior delta activity increases positive RewP amplitudes.

**ERP PE sequence results are robust to biases in subjective reward expectations.** Reward expectations are as critical to RPEs as the outcome, and failure to account for differences in these predictions across paradigms contributes to the disagreements in the reward EEG literature. For example, comparing ERPs following easy and hard neutral trials with identical feedback but opposite reward expectations showed FRN latency shifts according to RPE valence that matched those observed in losses and wins, respectively (see Supplementary Note 1). Furthermore,

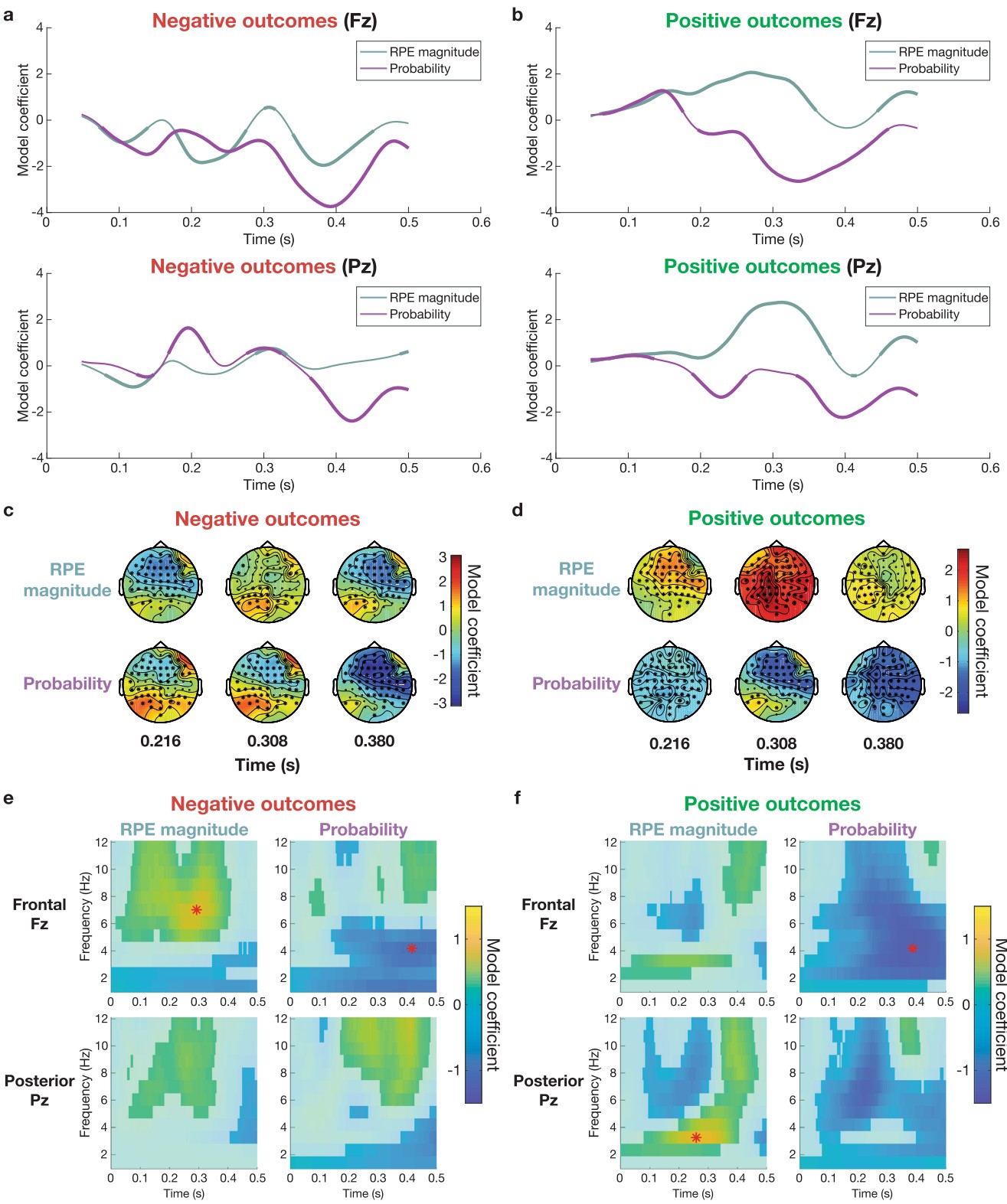

previous work has indicated participants' subjective reward expectations may deviate from objective probabilities established by the experimental design or modeled from behavior[39,70]. To assess whether this issue impacted our results, we collected an additional behavioral dataset to measure subjective ratings of reward expectations before feedback (see Methods). Ratings tracked difficulty (easy/hard) and trial outcomes (win/loss) and

revealed subjective biases such that participants underestimated their probability of winning in easy conditions and overestimated it in hard conditions (Supplementary Note 2 and Supplementary Fig. 10). The EEG results in Fig. 2 were reproduced when incorporating similar biases into our RL model, providing evidence that our conclusions are robust to differences between subjective and model-based reward expectations.

**Fig. 6 Separating outcomes by valence disentangles overlapping ERP and time-frequency power signatures of prediction errors.** Left column is for negative outcomes (easy loss, easy neutral, and hard loss), while right column shows results for positive outcomes (easy win, hard win, and hard neutral). **a** Model coefficients for effects of RPE magnitude and probability on ERP amplitude in only negative outcomes at frontal site Fz (top) and posterior site Pz (bottom). Bolding indicates significant timepoints ($q_{FDR} < 0.05$; $n = 32$ participants). ERP amplitude significantly decreases with RPE magnitude in the early FRN window only at Fz, and this effect fluctuates rhythmically to rebound and peak at a later epoch. **b** Same for only positive outcomes. RPE magnitude shows a different pattern than on negative outcomes, with a significant positive relationship with ERP amplitude peaking in the RewP/P3 window and at posterior Pz. In contrast, the late, frontal probability effect shows a consistent late, negative relationship with amplitude across positive and negative valence. **c** Single-trial regression over all 64 electrodes is computed for only negative outcomes in the same three 50 ms windows from Figs. 2 and 3. Stars indicate significant electrodes ($q_{FDR} < 0.05$; $n = 32$ participants). RPE magnitude shows significant negative effects in frontal sensors during the early and late windows, while probability shows a strong negative effect in fronto-central sites in the late window. **d** Same for only positive outcomes. The late, fronto-central negative effect of probability matches negative outcomes, but RPE magnitude effects are strongest in central and posterior sensors in the middle window. **e** Model coefficients fit to evoked power on negative outcomes for frontal electrode Fz and posterior electrode Pz, with nonsignificant points ($q_{FDR} > 0.05$; $n = 32$ participants) plotted opaquely. Red stars indicate maximal coefficients for each model predictor across both electrodes. RPE magnitude coefficients peak in frontal theta power, and probability coefficients peak later in frontal delta power. **f** Same for positive outcomes. The maximal RPE magnitude effect shifts to delta frequencies in posterior channels, indicating distinct mechanisms for RPE encoding on wins and losses. In contrast, probability effects maintain their late frontal delta distribution.

## Discussion

We tested competing valenced and non-valenced explanations of reward feedback-locked EEG signatures by separating overlapping ERP components and cognitive variables using single-trial behavioral modeling and multiple regression across temporal, spatial, and spectral dimensions. Analyses using the standard approach of combining wins and losses implied early frontal theta activity in the FRN epoch represented valenced, scalar RPE value, and subsequent posterior delta band activity at the P3 peak indexed non-valenced RPE magnitude, seemingly supporting a combination of classical RL and independent coding theories[35,43]. However, repeating the regression analyses for wins and losses separately revealed the early valenced RPE value effect was an artifact of overlap between two distinct non-valenced RPE magnitude effects, providing support for both the salience account[6,28] and a revised version of the RL theory[46,47]. Specifically, our data confirm recent studies arguing negative RPEs elicit a frontal negativity and theta power consistent with the FRN[62,63,74,81,82], while positive RPEs elicit a slower, ramping positivity in delta frequencies consistent with the RewP[46,50,51,53,61,63,69]. Correlations with benchmark ERPs from an Oddball task in the same participants highlighted mixed contributions of both N2-like and P3-like components in the window of the original valenced RPE value effect, consistent with FRN and RewP overlap. In contrast, we also observed a novel later fronto-central positivity on the downslope of the P3 that tracks outcome probability, was stable across wins and losses, and had no corresponding Oddball ERP. Notably, traditional mean window and peak-to-peak ERP metrics were less reliable than single-trial modeling and failed to disentangle these overlapping components and their relationships to PEs. Finally, model comparisons showed PEs captured EEG features better than outcome properties, and reward expectations modulated FRN latency on neutral trials, emphasizing that EEG signatures of reward processing are best viewed through the lens of predictive coding. Below, we discuss how our analysis strategy addresses core theoretical and measurement issues in the literature and the implications of our findings on the nature of reward processing EEG components and their proposed relationships to learning signals.

**Methodological implications.** Methodologically, our behavioral modeling and multi-dimensional regression approach improves on traditional ERP analyses in several ways. In terms of theory, many reward EEG studies use categorical ANOVA statistics[40,79,82] and test ERP sensitivity to experimental manipulations of outcome valence, magnitude, and probability that indirectly reflect hypothesized PE computations[49,51,52]. Instead, we directly test the central tenets of the main proposals in the field by combining single-trial estimates of RPEs derived from individual participant behavior into mixed-effects multiple regression analyses providing several advantages. Regression analyses incorporate directional hypotheses with continuous instead of categorical variables, produce signed model coefficients that obviate the need for post-hoc tests, and provide flexibility to analyze negative and positive outcomes separately and avoid confounding the overlapping FRN and RewP in a win-loss difference wave contrast. In contrast to single variable correlations, multiple regression partitions variance appropriately between the model's competing valenced and non-valenced predictors in a single analysis. Including random intercepts for each subject in a mixed-effects model also enhances statistical power to allow high-resolution analyses[72,73]. Importantly, regression frameworks provide formal model comparisons that quantitatively determined that EEG signatures of reward feedback are better described by modeling latent cognitive variables in a predictive coding framework[65,75,76] than by the experimentally manipulated outcome features used in standard analyses[43,51,52].

Regarding ERP measurement, single-trial regression at each time, electrode, and time-frequency point provides the high spatio-tempo-spectral resolution needed to disentangle multiple overlapping components[17,83,84]. In contrast, traditional mean window and peak-to-peak metrics provide a single measurement influenced by mixtures of components, usually averaged across trials and tested at the group level[38]. For the FRN window, these traditional methods replicated the valenced RPE value regression result, but also found significant non-valenced effects of RPE magnitude or probability depending on the metric, suggesting potential confounds from overlapping components. Notably, these non-valenced effects using traditional metrics were the only results that did not replicate across both cohorts, potentially due to their lower statistical power relative to the single-trial mixed-effects modeling. Also, mean window and peak-to-peak measurements are usually aligned to observable peaks, a strategy that would have failed to detect the late probability effect. In sum, these results highlight the benefit of unbiased, high-resolution, multi-dimensional regression analyses for separating and interpreting overlapping components.

**Theoretical implications: Predictive coding.** Our results emphasize the importance of accounting for reward expectations when interpreting reward processing EEG. Quantitative model comparisons showed a model comprised of outcome value, magnitude, and probability—features commonly associated with the FRN and RewP but that do not account for expectations—performed worse than the RL-based model across all ERP

features. The outcome-based model was improved when outcome value was replaced with outcome valence to account for different reward expectations on neutral trials in easy and hard conditions, although this outcome valence model was still worse than the RL model based on PEs. The importance of predictive coding was especially apparent on neutral trials with identical outcomes but different feedback valence based on expectations. FRN peak latencies were modulated by RPE value, and similar to losses and wins, FRN peak latency was earlier for neutral feedback with negative than positive RPEs. However, Williams et al. found an opposite FRN latency shift with wins occurring earlier than losses in a large ($n = 500$) gambling dataset using only visual feedback[54]. The latency of FRN and RewP ERPs in our data are also earlier than reported in recent meta-analyses[38], likely because feedback in our task includes auditory components, which generate faster FRN latencies[33]. These discrepancies and the presence of multiple overlapping components indicate that although our latency results provide evidence of the influence of reward predictions on neutral trials, they should be interpreted with caution and may not generalize to paradigms with different feedback stimuli or task demands[58,60,85].

In contrast to these different brain responses to neutral outcomes, post-experiment survey data indicated participants' explicit interpretations of neutral feedback did not differ across easy and hard conditions. This finding supports assertions that the FRN is generated by a habitual, model-free reward learning circuit that bypasses conscious representations of goal-directed task structure and instead relies on implicit associative learning mechanisms[37,71]. Nonetheless, previous work indicates that failure to account for subjective expectations can obscure reward EEG effects and confound their interpretation[70]. To address this issue, we collected subjective ratings of win probabilities in an additional behavioral experiment that showed participants' subjective expectations tracked our difficulty manipulation and their likelihood of winning or losing. We employed a control RL model incorporating subjective biases representative of those ratings to eliminate concerns that differences between subjective and model-based reward expectations influenced our EEG conclusions. However, these behavioral data and modeling analyses do not account for differences in subjective biases across participants or predictions on individual trials (e.g., accurately identifying errors on easy trials before feedback). Future studies may define how these effects modulate EEG signatures of PEs.

**Theoretical implications: ERP components**. Taken together, these observations clarify core issues in human reward processing EEG, confirming and extending recent proposals by disentangling the sequence of scalp ERP and TFR signatures unfolding after feedback and assessing their relationships to various cognitive PEs and canonical N2 and P3 ERPs. We employed mean window, peak-to-peak, and high-resolution regression analyses across wins and losses aiming to replicate commonly used win-loss difference waves[38,46,60]. Our results support a combination of the early, classical RL and independent coding theories, mainly that the FRN window represents a scalar, quantitative RPE instead of binary valence and the P3 represents non-valenced RPE magnitude. The overlapping epochs of significance for valenced RPE value and non-valenced RPE magnitude and probability predictors may also explain why some studies find no difference of valence and instead support the salience account, particularly since mean window and peak-to-peak metrics vulnerable to P3 confounds introduce significant non-valenced effects in our data. However, these approaches failed to capture the true nature of these components.

As suggested by the original authors of the RL theory, interpretations of any analysis that fails to separate wins and losses are flawed because they confound the FRN and RewP contributions on negative and positive outcomes, respectively[47,55,63]. Instead, temporal and spatial overlap of two distinct quantitative but non-valenced RPE magnitude effects in frontal theta on losses and posterior delta on wins creates the appearance of a valenced RPE value effect. Note that symmetric deflections in opposite directions caused by the FRN and RewP could potentially cancel out the quantitative tracking of RPE magnitude and explain why some authors observed only binary valence in this time window[36,43–45].

Cavanagh et al. and others have argued the frontal theta response underlying the FRN is an instance of a general MPFC control mechanism elicited by feedback that requires adaptation[28,29,67,78]. Time estimation tasks like ours employ an implicit win-stay lose-switch strategy in which control adjustments are needed after losses but not wins. Consequently, the FRN mainly appears following negative outcomes, although we do observe weak FRN-like deflections after positive feedback, potentially because vectoral RT feedback allows adjustments of motor timing even when previous responses were correct (e.g., when the target zone is large on easy trials). Importantly, this argument can also explain evidence supporting the salience theory based on paradigm differences in which conditions elicit non-valenced control PEs that trigger MPFC theta responses[6,28]. For example, FRN and theta responses have been observed following surprising positive outcomes in probabilistic learning tasks when unexpected rewards may indicate shifting reward contingencies that require modifying decision policies, thereby dissociating control PEs from negative valence[29,41,42,65,66]. These results suggest the FRN is not specific to negative RPEs as proposed by the original RL theory but is instead better described by a version of the salience theory in line with action-outcome PEs from the PRO model[6].

An outstanding question in the reward processing EEG literature is the nature of the RewP and other P3-like components. Our data substantiate claims that RPE magnitude on positive outcomes modulates a delta frequency P3-like component matching the RewP[46–48,63,69]. This effect appeared strongest in posterior electrodes similar to the topography of a P3b when analyzing wins and losses together. However, when examining only positive outcomes, the RewP had a more central distribution closer to a P3a, matching previous reports[46,51,63,69]. Further, Oddball ERP comparisons showed activity in this epoch correlated more with the P3a than with the P3b.

Whether the RewP is specific to positive RPEs is complicated by the fronto-central evoked positivity following the FRN on negative outcomes that also increases with RPE magnitude (see Fig. 2 and Supplementary Fig. 9). Previous studies analyzing wins and losses together have interpreted these two positive peaks as the same P3 component representing non-valenced RPE magnitude in both conditions along the lines of the independent coding hypothesis[43,74,86]. However, our data show stronger theta than delta power representations of RPE magnitude at that time point for negative outcomes (see Fig. 6e), as well as rhythmic fluctuations at theta frequencies in frontal RPE magnitude coefficients (Fig. 6a). These observations fit with an alternative interpretation that this positive peak on negative trials is not a P3 but instead due to phase reversal of the theta rhythm underlying the FRN[53,64]. This later interpretation would also explain why the RPE magnitude effect is stronger for losses in frontal sensors where theta is maximal but stronger for wins at posterior sensors where delta is maximal. Importantly, if the FRN and the following positivity are in fact generated by the same phase reset in MPFC theta activity—which is a more parsimonious explanation—they

should not be interpreted as unique components representing different aspects of PEs[56–58]. These observations also imply the theta response on negative trials may mask the RewP in commonly used difference wave contrasts. Notably, when the RewP is defined as the peak of the win-loss difference wave, it is maximal on the upslope of the P3 and at Cz[54], which are the precise points of maximal spatial and temporal overlap between FRN and P3 components. Taken together, these data suggest the RPE magnitude effect in our data is not a single non-valenced P3 as posited by the independent coding hypothesis but is due to spatio-temporal overlap between the P3-like RewP specific to positive RPEs and the positivity generated by phase reversal of FRN-linked theta tracking negative RPEs[53,54,62].

Our experimental design and modeling approach dissociated outcome probability as a second source of non-valenced salience, which has recently been shown to modulate DA coding of RPEs in monkeys[87]. Our multiple regression analyses revealed this predictor captured a late frontal positivity in delta frequencies with no observable peak in grand-averaged ERPs and no direct correlation with any Oddball ERPs, which to the best of our knowledge has not been described previously. The latency of this late frontal positivity matches the timing of a late probability effect identified in the meta-analysis reported by Sambrook & Goslin[38], but its relationship to previously reported ERPs is unclear. Its anterior topography and sensitivity to relatively novel feedback fit the description of the classic P3a[26], but previous studies have ascribed the P3a label to the positivity following the FRN[74,86]. Also, frontal P3a ERPs typically occur before posterior P3b ERPs, which contradicts the order of posterior RPE magnitude and anterior probability positivities in our results. Furthermore, this effect did not correlate with the novelty P3a from the Oddball task, although it is possible other components such as the negativity driven by the second cycle of frontal theta may have obscured this relationship. Alternatively, our late probability effect could be related the late positive potential (LPP), a positive ramping ERP that starts ~400–600 ms after feedback and is enhanced by motivational importance, but this seems unlikely given the LPP's posterior distribution and sustained time course of up to several seconds[88]. It is also possible that this effect is an artifact of correlations between RPE magnitude and probability in our model, but its replication across cohorts and in wins and losses separately, as well as the dissociable spatio-temporal patterns of model performance when excluding either source of salience argue against this possibility. Ultimately, this late fronto-central probability effect may correspond to a P3a-like ERP with altered timing due to the specific design of our task, but this finding should be replicated using different paradigms.

Understanding the nature of these different components is important for maximizing their potential in clinical applications. For example, in a population with comorbid anxiety and depression symptoms performing a probabilistic learning task, theta power on losses correlated with anxiety subscores while delta power on wins predicted depression, highlighting the dissociable relationships between these EEG signatures and correlated dimensions of psychopathology[63]. Strong links between DA and reward circuits and the RewP make it a promising biomarker for mood disorders and addiction[46,89,90]. DA markers predict personality traits like extraversion and sensation seeking[91,92], as well as psychiatric risk for schizophrenia[93], mood disorders[94], and addiction[95]. RewP amplitude also predicts extraversion[96], depression symptoms[97,98] and onset[99], substance misuse[100], and may mediate the relationship between DA and aberrant reward sensitivity in these disorders[101]. In contrast, components in the family of ERPs driven by mid-frontal theta power (e.g., N2, ERN, FRN) predict anxiety symptoms across multiple psychiatric

conditions[102]. These theta signals are hypothesized to reflect enlarged control PEs due to abnormal sensitivity to threat, reward, and punishment and hyperactive performance monitoring, resulting in more cautious and avoidant behavior[103,104]. Finally, P3 abnormalities are also observed in a host of disabling neuropsychiatric disorders including substance use disorders[105], bipolar disorder[106], and schizophrenia[107]. Given the complex relationships between these overlapping components and dissociable dimensions of clinical symptoms, refining specific mappings between reward and control PEs and EEG features across multiple dimensions could enhance the power and reliability of these biomarkers and improve their diagnostic and therapeutic potential.

In conclusion, our experimental design, computational modeling, and signal analysis approach provide a comprehensive assessment of the sequence of EEG components elicited by reward feedback and their relationships to control and reward PEs. Multiple regression analyses across temporal, spatial, and frequency dimensions of the data and correlations with canonical N2 and P3 ERPs from the Oddball task elucidate a succession of overlapping components, each corresponding to distinct PEs. We demonstrate the pitfalls of using standard mean window, peak-to-peak, and win-loss difference wave techniques that confound the early, frontal, theta frequency FRN tracking non-valenced RPE magnitude on negative trials and the concurrent ramping of more posterior delta frequency RewP responses driven by RPE magnitude on positive outcomes. Separating positive and negative outcomes and distinguishing temporal, spatial, and frequency dimensions confirmed an updated version of the salience account of the FRN for negative RPEs that concurs with the PRO model, provided evidence that positive RPEs elicit a P3-like RewP, and identified a novel late frontal P3 tracking low probability outcomes. In summary, we used traditional analyses contrasting wins and losses to reproduce classical evidence of valenced RPE value effects in the FRN window and non-valence RPE magnitude effects in the P3 window that formed the foundations of early RL and independent coding theories. However, follow-up analyses separating wins and losses revealed those interpretations were confounded by overlap of three distinct non-valenced salience components. Instead, our data corroborate and extend modern accounts of the FRN as an instance of control PEs generated by MPFC theta and of the RewP as a P3-like component tracking positive RPEs. Our findings demonstrate the power of behavioral modeling and single-trial EEG regression to separate overlapping components, adjudicate longstanding theoretical debates, and improve the utility of potential ERP biomarkers for diagnosis and treatment of neuropsychiatric disorders.

## Methods

**Experimental model and participant details**. Target Time EEG data were collected from 41 adult healthy participants (mean ± SD [range]: 20.5 ± 1.4 [18–25] years old; 28 women; 37 right-handed) at the University of California, Berkeley. Oddball EEG data were collected during the same session from a subset of 30 of these participants, 19 of which were in the replication cohort. A separate cohort of 24 healthy adults (mean ± SD [range]: 30.6 ± 5.3 [21–44] years old; 12 women; 22 right-handed) completed a follow-up remote behavioral version of the Target Time task to obtain subjective ratings of win probability. All participants reported no history of psychiatric or neurological disorders and had normal, or corrected-to-normal, vision. All participants were either financially compensated or given course credit and gave written informed consent to experimental protocols approved by the University of California, Berkeley Committees on Human Research.

### Method details

*Target Time behavioral task*. The Target Time interval timing task was written in PsychoPy[108] (v1.85.3) and consisted of eight blocks (four easy and four hard in randomized order) of 75 trials. Following central fixation for an intertrial interval randomly chosen as 700 or 1000 ms (an earlier task design also included 200 and 400 ms ITIs for four participants in the initial cohort), trials began with presentation of counter-clockwise visual motion from the bottom of a ring of dots at a

constant speed to complete the circle at the 1 second temporal interval. Participants estimated the interval via button press using an RTBox (v5/6) response device[109]. The width of a gray target zone indicated the tolerance for successful responses. Veridical win/loss feedback was presented from 1800–2800 ms and composed of (1) the tolerance cue turning green/red, (2) cash register/descending tones auditory cues, and (3) a black tick mark denoting the response time (RT) on the ring. Participants received ±100 points for wins/losses. Tolerance was bounded at ±15–400 ms, and separate staircase algorithms for easy and hard blocks adjusted tolerance by −3/+12 and −12/+3 ms following wins/losses, respectively. Participants learned the interval in five initial training trials in which visual motion completed the full circle. For all subsequent trials, dot motion halted after 400 ms to prevent visuo-motor integration, forcing participants to rely on external feedback. Training concluded with 15 easy and 15 hard trials to initialize both staircase algorithms to individual performance levels. Main task blocks introduced neutral outcomes on a random 12% of trials that consisted of blue target zone feedback, a novel oddball auditory stimulus, no RT marker, and no score change.

*Oddball behavioral task and performance.* The three-tone Oddball target detection task was written in PsychoPy[108] (v1.85.3) and consisted of 10 training trials followed by three blocks of 130 trials each. Following central fixation for an intertrial interval randomly chosen as 1.3 or 1.5 s, participants were presented with either a standard (75.5% of trials), target (12.25% of trials), or novel (12.25% of trials) audiovisual stimulus for 0.2 seconds. Participants were required to press a button using an RTBox (v5/6) response device[109] when they detected a target stimulus. The visual stimulus was the same as the feedback stimulus in the Target Time task —a ring of dots with a colored bar for the target zone—except without the black tick mark indicating response time and the green, red, and blue colors of the target zone were randomized to the standard, target, and novel conditions across participants. The accompanying auditory stimulus was a 440 Hz tone for standards, 1760 Hz tone for targets, and a novel, randomly selected oddball stimulus for the novel condition (different from Target Time neutral stimuli). Group-level accuracy was 99.68 ± 0.0061% (mean ± SD), and reaction times were 0.364 ± 0.063 s (mean ± SD).

*Post-experiment behavioral survey.* Immediately following the EEG experiment, 33 participants (n = 24 remain after all exclusion criteria, see Methods) were given a six-question survey to assess their interpretation of example pictures of winning, neutral, or losing outcomes with small or large target zones to indicate easy or hard contexts. This data are only available in a subset of participants because data collection began three participants before implementation of the Oddball task. In response to the question "How would you feel about this feedback?", participants rated each outcome on a 9-point Likert scale, where 1 indicated "Terrible!", 5 indicated "I don't care…", and 9 indicated "Great!". Answers are reported after centering the ratings at the indifference point of 5.

*Subjective ratings target time task.* In a follow-up remote behavioral experiment, participants downloaded and completed a version of the Target Time task optimized to assess subjective ratings of win probability. In this version of the task, participants completed six blocks of 75 trials after the same 35 training trials and responded using the mouse. Importantly, after responding but before feedback, participants were asked on every third trial to rate "How likely is it that you won on this trial by responding in the target zone?" and responded by clicking on a slider bar rating scale between two "0" and "100" tick marks at the far left and far right labeled "Definitely Lost" and "Definitely Won", respectively. Participants were given 20 seconds to respond before the rating timed out and feedback was presented. Participants with prior knowledge of outcome probabilities in the task (n = 2) were excluded after preliminary analyses revealed qualitatively different (less variable) ratings.

*Behavioral modeling.* Target Time EEG participants were excluded because of technical recording errors (n = 4 datasets with missing EEG or metadata necessary for analysis), excessively noisy data (n = 2 datasets with >3 standard deviation outliers in number of epochs or timepoints rejected based on visual identification of large, global artifacts), or poor behavioral performance (n = 3 datasets where RT outlier exclusion criteria resulted in <20 trials in any condition), leaving 32 participants for analysis. All Target Time analyses were piloted on an initial cohort of 15 participants to finalize model parameters and statistical tests before results were replicated in a second cohort of 17 participants. All findings except point estimate results in Supplementary Fig. 3b–d successfully replicated across cohorts (see Supplementary Table 1 for differences in these results across cohorts), so all other results presented in the text and figures reflect all 32 datasets combined.

The relationship between the tolerance around the target interval and expected value was fit to individual participant behavior using logistic regression. Specifically, tolerance was used to predict binary win/loss outcomes across trials using the MATLAB function *glmfit* with a binomial distribution and logit linking function. Trials with neutral outcomes were excluded because they were delivered randomly and thus not reflective of performance. The probability of winning ($p_{win}$) for each participant was computed as:

$$p_{win} = \frac{1}{1 + e^{-(\beta_0 + \beta_1 t)}} \quad (1)$$

where $\beta_0$ is the intercept and $\beta_1$ is the slope from the logistic regression, and $t$ is the tolerance on a given trial. Expected value was derived by linearly scaling the probability of winning to the reward function ranging from −1 to 1. RPE value was then computed by subtracting expected value from the actual reward value, and RPE magnitude was computed as the absolute value of RPE value. Outcome probability was simply the proportion of each outcome across easy and hard blocks separately. See Supplementary Fig. 1 for model predictions by condition.

Although RPE magnitude and probability predictors were correlated (r = −0.71), variance inflation factors, which measures the degree of collinearity, were $VIF_{RPEmag} = 2.0$ and $VIF_{Prob} = 2.0$, which is below even the most stringent recommended thresholds of 2.5 for excluding them from the same model[110]. Nonetheless, the separate contributions of these two predictors were assessed using versions of the RL model excluding each of these two predictors, and time-resolved model comparison and coefficient results are plotted in Fig. 2e,f and in Supplementary Fig. 4g–j, respectively (see below for details).

Notably, this model was fit across all blocks after training under the assumption that participants learned the task during the 35 training trials, and that the staircase algorithm was appropriately initialized to the participant's skill level in the training. Since our model is fit using behavior over the entire session, it is possible that it would not describe early trials well, especially if learning occurs over the course of the session. As control analyses, we computed expected value after replacing single-trial win probabilities with block-level accuracy, as well as a rolling average of accuracy on the last 5 or 10 trials. Our single-trial logistic regression model outperformed all of these control models (higher $R^2$ and lower AIC) for mean window, peak-to-peak, and single-trial amplitude regression analyses.

To compare our main model based on RL principles to models similar to those commonly used in the literature, we computed a similar model using only outcome features that did not account for reward expectations. The Outcome Value model included the value (−1, 0, and 1 for losses, neutral, and wins), magnitude (0 for neutral, 1 for wins and losses), and probability (same as above) for each outcome. The Outcome Valence model was identical to the Outcome Value model except the value predictor was replaced by a valence predictor that treats neutral trials as valenced reward omissions, meaning losses and easy neutral outcomes were coded as −1 and wins and hard neutral outcomes were coded as 1.

*Electrophysiology recording.* EEG data were recorded using a BioSemi ActiveTwo amplifier with a 64-channel active electrode system arranged according to the extended 10–20 system at a sampling rate of 512 Hz. Horizontal electrooculogram (EOG) were recorded from electrodes placed at both outer canthi, and vertical EOG were recorded from an electrode placed below the right eye and right front-topolar electrode FP2. Additionally, two external electrodes were placed on each ear lobe for use in offline re-referencing.

*Electrophysiology and behavior preprocessing.* Preprocessing and analysis used the Fieldtrip toolbox[111] and custom code in MATLAB and Python. EEG data were bandpass filtered from 0.1–30 Hz, demeaned, re-referenced to the average of both ear lobe channels, and then downsampled to 250 Hz. Excessively noisy epochs and channels were removed by visual inspection. Independent component analysis (ICA) was used to remove artifacts due to channel noise, muscle activity, heartbeat, and EOG (i.e., components correlated with bipolar derivations of horizontal or vertical EOG signals bandpass filtered from 1–15 Hz). Trials were segmented from −0.15 to 2.8 s relative to trial onset, and missing channels were interpolated from neighboring channels via Fieldtrip function *ft_channelrepair*. Final quality checks rejected trials for behavioral outliers (RTs missing, <0.6 s, or >1.4 s) or EEG artifacts including muscle activity, large voltage shifts, and amplifier saturation identified via visual inspection and using the Fieldtrip function *ft_reject_visual*, resulting in trial counts ranging from 448–524 (mean ± SD: 498.4 ± 20.1). In the remote behavioral subjective rating task, outliers were rejected for the same interval timing response RT criteria and for slow subjective ratings with RTs greater than three standard deviations from the mean, resulting in 382–450 trials (mean ± SD: 437.1 ± 17.6) and 125–150 ratings per participant (mean ± SD: 143.5 ± 6.2). Oddball EEG preprocessing was identical except trials were initially segmented from −0.2 to 1.3 s, and RTs were rejected as outliers if less than 0.1 s or greater than 1.3 s, resulting in trial counts ranging from 334-389 (mean ± SD: 379.5 ± 10.9).

*Event-related potentials and difference waves.* EEG data were realigned to feedback onset and cut to −0.2 to 1 s. ERPs were calculated for each participant by bandpass filtering from 0.5–20 Hz, baseline corrected by subtracting the mean of 200 ms immediately preceding feedback, and averaging across trials. Oddball ERPs were identical except aligned to stimulus onset.

Difference waves were computed to facilitate visual comparisons to previous FRN/RewP studies and to valence, magnitude, and probability effects from our model-based multiple regression results. All difference waves were computed at the individual level for both Fz and Pz and then plotted as grand-average waveforms with standard error of the mean across participants. The simplest RewP contrast was computed by subtracting the ERP averaged over all negative valence conditions (easy neutral, easy loss, and hard loss) from the ERP averaged overall positive valence conditions (easy win, hard neutral, and hard win). Outcome valence difference waves were also computed between condition pairs matched for outcome magnitude and probability: hard win minus easy loss (large magnitude,

low probability), easy win minus hard loss (large magnitude, high probability), and hard neutral minus easy neutral (small magnitude, low probability). Outcome magnitude difference waves were computed by subtracting small from large magnitude outcomes matched for valence and probability: easy loss minus easy neutral (negative valence, low probability) and hard win minus hard neutral (positive valence, low probability). Outcome probability difference waves were computed by subtracting likely from unlikely outcomes matched for valence and magnitude: easy neutral minus hard loss (negative valence, low magnitude) and hard neutral minus easy win (positive valence, low magnitude).

*ERP point estimates and latencies*. To facilitate comparisons with prior FRN/RewP studies, we computed traditional mean window and peak-to-peak point estimates of FRN amplitude at electrode Fz. The mean window metric was calculated as the mean amplitude of each participants' condition-averaged ERPs in a 100 ms window centered on that participant's FRN peak latency computed across all conditions. The peak-to-peak metric was calculated by subtracting the FRN peak amplitude from the amplitude of the preceding positivity (P2) for each condition and participant. To account for variability in ERP waveshapes at the single participant level, peak-to-peak amplitude was only computed if a positive peak was found in the interval 100–260 ms post-feedback that preceded a negative peak in the interval 180–300 ms. According to these criteria, peak-to-peak amplitude could not be reliably computed on 11/192 ERPs. Additionally, the latency of the negative peak in this analysis was used as the FRN peak latency, which was then normalized within participant by subtracting the mean latency across all conditions.

To aid interpretation of ERP features underlying model-based results, we computed reference ERPs in the Oddball EEG data quantified using the mean across 50 ms windows of condition-averaged participant ERPs. For the Novel N2b and Target N2c, mean windows were centered on the peak negativity from the grand-average ERP at Fz between 0.2 and 0.3 s in their respective conditions. For the Novel P3a and Target P3b, mean windows were centered on the peak positivity from the grand-average ERP between 0.3 and 0.45 s at Cz and Pz, respectively.

*Time-frequency representations*. EEG data were realigned and segmented from −0.2 to 1 s around feedback onset. Spectral decompositions were estimated at each time point by convolving the signal with a set of complex Morlet wavelets, defined as complex sine waves tapered by a Gaussian. The frequencies of the wavelets ranged from 1–12 Hz in 1 Hz linear steps. The full-width at half-maximum (FWHM) ranged from 1.184–0.096 s with increasing wavelet peak frequency, which corresponds to three cycles per frequency. Task-evoked power was computed as the square of the magnitude of complex Fourier-spectra and baseline corrected by decibel conversion relative to a 200 ms baseline immediately preceding feedback.

### Statistics and reproducibility

*Time-resolved modeling*. We adopted a multiple linear regression framework to directly compare the predictive power of valenced and non-valenced PEs derived from our RL-based behavioral model and simple outcome features that do not account for expectations. Our full RL model combines our single-trial model estimates of expected value, valenced RPE value, non-valenced RPE magnitude, and outcome probability in a linear mixed-effects model with random intercepts for each participant to maximize statistical power by accounting for within participant variance. This model was used to predict the temporal evolution of EEG amplitude at each time point from 50 to 500 ms post-feedback using the MATLAB function *fitlme*, which tests significance of model coefficients using two-sided *t*-tests under the null hypothesis the coefficient is equal to zero. Resulting *p*-values were corrected for multiple comparisons using false discovery rate[112] across timepoints and model predictors. For clarity, any *p*-values corrected for multiple comparisons are reported as $q_{FDR}$ throughout the manuscript. These analyses were run separately for electrodes Fz and Pz to assess frontal FRN and posterior P3 ERPs.

To compare the performance of our RL-based and outcome-based models, we ran the same time-resolved regression analyses using Outcome Value and Outcome Valence models. Model performance was quantified using the Akaike Information Criterion (AIC), which scores model performance based on variance explained while penalizing models with extra parameters. Lower AIC values indicate better model performance. To emphasize differences between models, AIC is reported relative to a baseline model containing only random intercepts for each participant, which is equivalent to the mean ERP across all conditions. Finally, the relative contributions RPE magnitude or probability to predicting ERP amplitude in the P3 window are assessed by reporting AIC of the RL model when leaving out either of these two correlated predictors.

*FRN and P3 point estimate modeling*. This modeling procedure was also used to predict mean window and peak-to-peak measurements of FRN amplitude at electrode Fz and mean window P3 amplitude at Pz. Since these metrics yield one value per condition per participant, each model predictor was averaged within condition for each participant. FDR corrections were applied across model predictors. AIC is reported for this procedure to compare the RL model with Outcome Value and Outcome Valence models in the FRN point estimates and the RL model

with and without RPE magnitude and probability predictors in the P3 mean window analysis. A nearly identical multiple regression analysis was used to predict FRN peak latency, except the MATLAB function *fitglm* was used without the random intercept for participants because peak latencies were already normalized within participant. A two-sided paired samples *t*-test was used to test whether FRN peak latencies were different between neutral feedback in easy and hard conditions.

*Topography modeling*. To examine the spatial distribution of PE effects on evoked potentials, ERP amplitudes were averaged for all electrodes in three 50 ms windows centered on the largest coefficient from the time-resolved regression for RPE value (216 ms at Fz), RPE magnitude (308 ms at Pz), and outcome probability (380 ms at Fz). The multiple regression model was then used to predict amplitude at each channel in each window, and FDR corrections were applied across all channels, model predictors, and windows.

*Time-frequency power modeling*. Time-frequency representations were analyzed using the same mixed-effects multiple linear regression model to predict evoked power at each time-frequency point from 0 to 500 ms and 1–12 Hz. FDR multiple comparison corrections were applied across timepoints, frequencies, and model predictors, again separately for Fz and Pz to assess frontal FRN and posterior effects. These analyses were repeated for only negative and only positive outcomes to test whether these results were driven by feedback of one particular valence.

*Oddball-Target Time ERP correlations*. Post-hoc exploratory analyses compared Target Time ERP amplitudes in epochs showing maximal RL model-based effects to canonical Oddball ERPs via inter-participant correlations. These analyses were conducted in the subset of participants with both Target Time and Oddball EEG data (n = 22) after excluding one participant for excessive number of trials rejected due to noise in the Oddball EEG data and another participant for outlier Oddball behavioral accuracy (both outliers >3 standard deviations from the group mean). For each Target Time condition, individual participant amplitudes of Novel N2b, Novel P3a, Target N2c, and Target P3b ERPs were used as benchmarks of participant's N2/P3 amplitudes and correlated with the 50 ms mean window amplitudes in three epochs used for topography modeling: one at Fz centered on the peak RPE value effect (216 ms), one at Pz centered on the peak RPE magnitude effect (308 ms), and one at Fz centered on the peak Probability effect (380 ms). Correlation *p*-values were FDR corrected for the number of Oddball ERPs and the number of Target Time conditions.

*Post-experiment survey ratings*. Subjective ratings for neutral trials were tested for significant differences from the indifference point on the 9-point Likert scale after subtracting 5 to center ratings, separately for easy and hard trials. Rating data were tested using two-sided independent samples *t*-tests under the null hypothesis that ratings were from a normal distribution with mean equal to zero.

*Remote behavioral task subjective ratings*. We evaluated whether objective win probabilities derived from behavior deviated from participants' subjective experiences of reward probabilities. To accomplish this, we compared our measure of expected value computed via logistic regression of observed wins and losses to subjective ratings of win probabilities measured in the remote behavioral Target Time task. We correlated single-trial model-based win probability with subjective ratings across all participants and conditions, as well as independently for easy and hard conditions. For each participant, subjective bias was quantified separately for easy and hard conditions by the mean difference between subjective ratings and model-based win probabilities. To test whether subjective ratings were sensitive to the probability of winning or losing, ratings were z-scored within each participant and condition. Two-sided independent samples *t*-tests were used to compare normalized ratings before wins and before losses using group aggregated data for only easy, only hard, and both easy and hard conditions combined.

**Reporting summary**. Further information on research design is available in the Nature Research Reporting Summary linked to this article.

## Data availability

The datasets generated and/or analyzed during the current study are available in the Open Science Foundation repository and can be found at https://doi.org/10.17605/OSF.IO/JGXFR (ref. [113]).

## Code availability

Custom Python and MATLAB code used for preprocessing and analysis is available as a GitHub repository (https://github.com/hoycw/PRJ_Error_eeg), which includes system requirements and dependencies.

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

## Acknowledgements
We thank I. Griffith for help piloting the paradigm and A. Shah and J. Abbas for help collecting the data. This work was supported by NINDS R37NS21135 (RTK), CONTE Center PO MH109429 (RTK), NSF GRFP (CWH), and Greater Good Science Center Fellowship (CWH).

## Author contributions
C.W.H. and R.T.K. designed the experiment. C.W.H. and S.C.S. collected the data. C.W.H. and S.C.S. analyzed the data. C.W.H. and R.T.K. wrote the paper.

## Competing interests
The authors declare no competing interests.
