## [Peer Review File · Communications Biology]

Reviewers' comments:

Reviewer #1 (Remarks to the Author):

In their manuscript entitled "Multiple sequential prediction errors during reward processing in the human brain", Hoy and colleagues present scalp EEG data collected during a version of the interval timing task that was carefully designed to disentangle surprise magnitude (unsigned prediction error), surprise valence (reward prediction error), and outcome probability. Using trial-to-trial modeling of the EEG response to trial outcomes, the authors find that RPE maps onto the FRN, an surprise maps onto a subsequent P300 waveform, and trial outcome maps onto a subsequent late frontal positivity.

Overall, this is a very well conducted study. The task is nicely designed, the analyses are state-of-the-art, and the trial-to-trial aspect in particular can reveal insights into neural dynamics that are near-impossible to obtain using averaging-based methods. In particular, the fact that the single-trial modeling can reveal and resolve overlap between the two main components of interest is a very powerful demonstration of this approach.

My main point of criticism is that as a (somewhat informed) reader, it was somewhat difficult to me to determine what we learn from the current experiment that we did not already know (or knew to hypothesize) based on the Yeung & Sanfey paper from 2004 (J Neuro, cited in the manuscript). In essence, the current paper is a really well done replication that uses a single-trial approach to arrive at the same conclusion: the FRN reflects valence, the posterior P300 does not. Beyond the superiority of the single-trial method, there are some obvious additional aspects of the current paper that are worth mentioning – the finding regarding the late frontal positivity being one, the time-frequency results being the other. However, the paper is mainly couched in the context of reinforcement learning theory and its predictions vis-à-vis the FRN and P300 waves. In terms of conceptual knowledge, therefore, I think the advance is somewhat limited. At a minimum, the authors should perhaps focus a more substantial chunk of their paper on outlining the implications of their findings that go beyond Yeung & Sanfey.

The second rather notable shortcoming is that there is a substantial amount of literature that uses the single-trial EEG approach employed by the current authors to investigate surprise and/or reinforcement learning. Much of that literature is not discussed (or citted). Some examples of papers that investigated the relationship between P300 waveforms and surprise are Seer et al. (Brain & Cognition, 2016), Mars et al. (J Neuro 2008), Wessel et al. (PLoS Comp Biology 2019). In the domain of the FRN, both Becker et al. (J Neuro 2014) and Meadows et al. (Biological Psychology 2016) have used single-trial analyses, and should likely be cited. So should Talmi et al. (J Neuro 2013), as a major example of a study that doesn't use single-trial EEG, but whose results cast some doubt upon the proposed interpretation of the FRN (and that of Yeung and Sanfey). While perhaps not all of this literature needs to be included, a more comprehensive discussion that couches the current findings into the existing literature (beyond the basics of RL theory) should be undertaken.

Minor points

- Why was the behavioral survey only given to a subset of subjects?
- What criteria were used to evaluate sufficient data quality and behavioral performance to warrant inclusion/exclusion?
- The Morlet wavelet for the time-frequency analysis should be described in more detail (cf., MX Cohen, NeuroImage 2019).

Reviewer #2 (Remarks to the Author):

In this manuscript, the authors aim to correlate different types of information content with scalp-

recorded EEG signals specific to reward and punishment outcomes. The intro proceeds as an interesting and well-informed tour through the field of event-related EEG and different types of prediction errors. However, some critical parts of the theoretical background are outdated and a lot of very important literature is missing. I focus my review on these critical interpretative features and don't focus much on the methods themselves, which are of secondary importance in regards to the conclusions that could be drawn from this task and approach.

Major Points

1) In general, the account of "RL theory" of the FRN here is outdated. This is critical, as the setup (lines 73-75) and conclusion that the original DA hypothesis needs revision (lines 346-348) are not relevant logical foundations. This theoretical revision was long ago suggested by Holroyd himself (PubMed ID: 18513364), where he suggested that the FRN difference wave was due to a generic novelty N2 and a reward-specific component, now called the Reward Positivity. This is of course complicated by the fact that some studies refer to the difference wave as the FRN or as the RewP, whereas other studies use these terms to describe the condition-specific ERP component (FRN=N2, Reward evokes a RewP alone). Given this messy state of affairs, the authors should clarify their terminologies. However, this raises other subtle, but critical issues as well. The punishment-evoked FRN (or N2 if you want to call it that) does indeed correlate with negative RPE (20858518, 31149639, others, and so does theta power), but this does not mean that it only encodes this information quantity. In short, FRN and theta are sensitive but not specific to RPE (17850238, 18513364, 22120491, although this becomes complicated when using a difference contrast). In contrast, the reward-specific RewP (i.e. not the difference wave), seems to encode for positive RPE only (25676913), being both sensitive and specific to this information quantity. In sum, the FRN/N2 (to punishments) and RewP (to rewards) appear to be different signals from different systems that do different things. Current theories suggest such complexities that are indirectly related to the original RL theory (18513364, 25676913), so how does this latest state of affairs influence the author's theoretical perspective?

2) The authors need to demonstrate how this work is novel. Point-by-point regressions with unsigned PE and RPE can be seen in many prior studies, for a few examples see work by Philiastides (particularly in regard to signed vs. unsigned PEs evolving over time, e.g. 20510376), Fischer (particularly in regard to the topographic displays, e.g. 24050408), Cavanagh (31149639), Collins (29463751), etc.

3) One assumption that this paper makes that is distressingly very common in the field is to consider different ERP components as separate entities. On pg. 5 (and re-raised throughout the results), everything that is said about the relationship between the FRN and the P300 could certainly be true. The FRN and P3 might be unique bumps of evoked activity. Alternatively, the transition from the peak negativity (FRN) up the slope to the next positivity (P300) could be due to overlapping phase-locked patterns (as could N1 to P2, P2 to N2, etc. see work by Klimesch: 17459593, 15749980), meaning they share at least some variance (likely changing over time). We really don't know the ground truth, but this is reason to be careful and evaluate all possible theoretical accounts.

4) The task appears to evoke a tacit win-stay lose-switch strategy, which is common amongst similar EEG tasks, causing the results to look the same across studies. In such tasks, punishments, but not rewards, require behavioral adaptation, invoking the midfrontal action monitoring system (e.g. 24835663). This diminishes the utility of this specific study for theoretical advancement, since the outcomes will likely look like other WS-LS studies. For example the conclusion drawn on lines 91-92: "This sequence of prediction errors confirms the RL theory of the FRN ..." is far too optimistic and in my view it is both theoretically and methodologically incorrect (see point #1 above). At best, the finding reported here provides a single piece of evidence for one specific theory. But it is a piece of evidence that seems to replicate prior WS-LS findings instead of revealing new ones, and this might be due to the fact that many studies are simply testing the same thing - evoking one type of finding

(correlation with signed RPE). The fact that the FRN responds to both signed RPE (citations above) and unsigned RPE (17892382, 21295109, 22120491) suggests that the reality is more complex, and different tasks need to be specifically designed to probe different theoretical accounts.

Signed, James F. Cavanagh

Reviewer #3 (Remarks to the Author):

The authors report results from a single experiment and replication aimed at distinguishing two prominent accounts of dACC/mPFC function - one based on salience, and one based on valenced reward prediction error. To answer this, they manipulate reward probability across separate blocks of a behavioral task in which subjects are required to respond within a visually-cued temporal window while recording EEG, against which they regress model-derived signals, including RPE and unvalenced RPE magnitude.

ERP analysis identifies significant valenced RPE effects associated with the Feedback-Related Negativity (FRN) followed by significant RPE Magnitude effects associated with the P300. The further find significant valenced RPE signatures in the theta band for Fz, and significant RPE magnitude in delta for Pz. Based on these analyses, the authors conclude that the valenced RPE account of the FRN is fundamentally correct, but in need of revision to incorporate a larger diversity of reward-signals.

mPFC function remains an ongoing topic of debate, especially w/r/t the role of the region in salience or surprise vs processing value information, and I appreciate the authors' efforts to quantitatively test these competing accounts. However, I have major reservations regarding the computational methods.

The authors model behavior (accuracy) with a logistic function to derive trial-by-trial Win probabilities. From there, they derive value-based quantities, including EV, RPE, and salience (absolute RPE). Although these values derive from the observed behavior of the subjects, they do not necessarily reflect the subjective beliefs about behavior. Subjects could be very bad at tracking the objective probabilities (e.g., Hajcak et al., 2007), and therefore the values derived from objective behavior would be irrelevant to the FRN.

Some models based on the salience account (such as the PRO model cited by the authors) explicitly address how differences in subjective estimates of probability, deriving from, e.g., relative sensitivity to reward or error feedback, could lead to changes in the magnitude of error feedback in mPFC. If, as in the Hajcak study, subjects in this study overestimated their performance in the HARD condition, it would be possible to observe unvalenced RPEs that correlate strongly with (objective) valenced RPEs. Since the main point of the manuscript is to adjudicate between the salience and RL theories, it is essential that the subjective probability estimates be used to calculate RPEs. I don't see how that is possible with this study, however.

Beyond this major concern, however, the manuscript is well-organized and clearly stated. If the issue with subjective vs objective probability can be resolved in a revision, the results presented would provide additional perspective to a long-running debate in the cognitive control literature.

Response to Reviewers:

We thank the reviewers and editorial office for their thoughtful comments, which strengthened the manuscript. Our main goal is to assess brain responses to reward contingencies, which are debated in the EEG community due to a combination of theoretical and methodological challenges. Resolving these issues is important because these neural signals are central to the ongoing translational drive for non-invasive biomarkers of anxiety, depression, addiction, and other neurological and psychiatric conditions.

The key issues raised by the reviewers were novelty, methodological and theoretical concerns related to our analyses and interpretations of the overlapping FRN and RewP ERPs, and potential confounds of differences between model-based and subjective reward expectations. We address each of these points below. Specifically, this revision now includes a novel frontal P3-like probability component; new analyses that reveal how classical valenced RPE findings derived from common win-loss contrasts are confounded by overlap of non-valenced FRN and RewP components on losses and wins, respectively; new model comparison analyses demonstrating the benefits of predictive coding frameworks that account for reward expectations; a new EEG dataset in n=22 of the same participants using Oddball ERP comparisons to quantify contributions of overlapping N2 and P3 components; and finally we address Reviewer #3's concerns using a newly acquired behavioral dataset (n=24) measuring subjective ratings of reward expectations, new analyses demonstrating strong correspondence between model-based and subjective estimates of reward probabilities, and new simulations confirming subjective biases do not impact our EEG modeling results. This combination of results building from traditional summary metrics to high-resolution single-trial modeling reconciles evidence supporting early and modern theoretical accounts and clarifies longstanding debates in the reward processing literature.

Please see below for a **point-by-point response in bold** to each *reviewer's comments, which are underlined in italics*. Please note that **new text added to the manuscript is highlighted in red, bold fonts**.

Reviewer #1:

In their manuscript entitled "Multiple sequential prediction errors during reward processing in the human brain", Hoy and colleagues present scalp EEG data collected during a version of the interval timing task that was carefully designed to disentangle surprise magnitude (unsigned prediction error), surprise valence (reward prediction error), and outcome probability. Using trial-to-trial modeling of the EEG response to trial outcomes, the authors find that RPE maps onto the FRN, and surprise maps onto a subsequent P300 waveform, and trial outcome maps onto a subsequent late frontal positivity.

Overall, this is a very well conducted study. The task is nicely designed, the analyses are state-of-the-art, and the trial- to-trial aspect in particular can reveal insights into neural dynamics

that are near-impossible to obtain using averaging- based methods. In particular, the fact that the single-trial modeling can reveal and resolve overlap between the two main components of interest is a very powerful demonstration of this approach.

We thank the reviewer for their appreciation of the study and for their helpful comments, and we agree that resolving overlapping components is an important prerequisite to interpreting reward processing EEG data.

My main point of criticism is that as a (somewhat informed) reader, it was somewhat difficult to me to determine what we learn from the current experiment that we did not already know (or knew to hypothesize) based on the Yeung & Sanfey paper from 2004 (J Neuro, cited in the manuscript). In essence, the current paper is a really well done replication that uses a single-trial approach to arrive at the same conclusion: the FRN reflects valence, the posterior P300 does not.

We agree that the original framing and results of the initial submission largely replicated the findings of Yeung & Sanfey (2004). Below we describe new data and analyses in our revisions that substantially shift the results and theoretical interpretations to refute elements of these classic proposals in favor of modern accounts. However, we also note that replication of the Yeung & Sanfey (2004) study is an important endeavor in its own right. Indeed, that study has been selected as one of the 27 most influential and continually cited EEG studies (with the 4th most citations among those; Pavlov et al. 2021 Cortex Fig. 4), and the #EEGManyLabs consortium will attempt to replicate these findings in 3 separate laboratories. We believe this project reemphasizes the importance and ongoing nature of this debate and the value of our attempts in the revision to clearly differentiate the components following reward feedback and their relationships to cognitive constructs of interest.

Nonetheless, our revisions include several new analyses and results with important implications for the theoretical debate in reward processing EEG. Please note the points below are salient to both Reviewer #1 and Reviewer #2's concerns.

- 1. We use model comparisons to highlight how outcome-based analyses fail to account for reward expectations, emphasizing the importance of predictive coding.**
- 2. We reproduce classic results supporting early RL and independent coding hypotheses using a comprehensive battery of traditional mean window, peak-to-peak, and multiple regression analyses across wins and losses (similar to common difference wave analyses, which are provided in the supplement). However, our new approaches show how interpretations based on standard win-loss contrasts confound the effects of multiple components by failing to separate the overlapping FRN and RewP components on losses and wins, respectively, which we address in detail in the revision and the points below.**
- 3. We address the critical issue of component overlap by quantifying contributions of N2-like and P3-like activity using correlations between reward activity and benchmark ERPs measured in an Oddball task performed by the same participants.**

4. We apply single-trial regression across temporal, spatial, and spectral dimensions separately for wins and losses to reveal the nature of three overlapping components. In particular, direct comparisons to results when analyzing wins and losses together show the valenced RPE value effect is an artifact of component overlap, which refutes the original RL theory claim that the FRN is a valenced RPE signal. Furthermore, our results suggest the positivity following the FRN on negative outcomes may be a consequence of phase reversal in theta activity generating the FRN, thereby invalidating the independent coding hypothesis claim from Yeung & Sanfey (2004) that a single P3 component represents non-valenced RPE magnitude across wins and losses.

Beyond the superiority of the single-trial method, there are some obvious additional aspects of the current paper that are worth mentioning – the finding regarding the late frontal positivity being one, the time-frequency results being the other. However, the paper is mainly couched in the context of reinforcement learning theory and its predictions vis-à-vis the FRN and P300 waves. In terms of conceptual knowledge, therefore, I think the advance is somewhat limited. At a minimum, the authors should perhaps focus a more substantial chunk of their paper on outlining the implications of their findings that go beyond Yeung & Sanfey.

We thank the reviewer for recognition of the value of our late frontal positivity and time-frequency results, and for their emphasis on articulating the novel theoretical contributions of our work. The framing of the paper has been extensively revised both conceptually and with new data and analyses as noted above, which we believe substantially improve the contributions of our study. Below we quote several places in the text that highlight these revisions.

Most importantly, reporting results with both separate and combined analyses of wins and losses reveals the pervasive problem of overlap between the FRN and RewP and can explain away the effect of valence that has remained prevalent in the field.

INTRODUCTION (lines 154-175): “We found that when modeling wins and losses together as done in standard analyses, early, frontal theta activity underlying the FRN is best described by valenced RPE value, while non-valenced RPE magnitude and probability effects drive later delta activity consistent with P3 ERPs. Model comparisons show these data are better explained by RL-based PEs than outcome features and confirm the importance of predictive coding principles. Mean window and peak-to-peak FRN analyses replicated these RPE value effects but also showed non-valenced effects and were unstable across the two cohorts. These results suggest the FRN represents a scalar, quantitative RPE while two P3 ERPs encode non-valenced RPE magnitude and outcome probability. However, these conventional analyses combining wins and losses confound the FRN on negative trials and the RewP on positive trials, and comparisons to oddball task ERPs suggest a mixture of N2 and P3 contributions to the RPE value window. Indeed, modeling EEG amplitude, topographies, and time-frequency power after separating wins and losses reveals that the valenced RPE value effect is an artifact of

non-valenced RPE magnitude driving two overlapping FRN and RewP components in theta and delta frequency respectively, while the late frontal probability effect in delta is stable across outcomes. Finally, we use subjective ratings obtained in a follow up behavioral experiment to confirm these EEG results cannot be explained by subjective biases in reward contingencies. Collectively, these results provide strong evidence that human EEG following reward feedback is composed of a sequence of multiple overlapping neurophysiological signatures best accounted for by specific PEs in a predictive coding framework.”

RESULTS (lines 324-334): “Collectively, these results characterize a cascade of multiple PEs unfolding during the FRN and RewP epochs in reward processing EEG, starting with an early, frontal, valenced RPE value signal in the FRN time window, followed by a later, posterior, non-valenced RPE magnitude effect in the P3 window, and finally a later, fronto-central probability effect. These findings replicate evidence supporting the RL account of the FRN as a scalar, quantitative RPE instead of binary valence³⁵⁻³⁷ and the independent coding proposal’s separation of early valenced effects in the FRN window from two later non-valenced P3 effects^{43,45}. We re-assess the interpretations of these effects after characterizing their spatial and frequency distributions, their correspondence with benchmark N2 and P3 ERPs from the oddball task, and most importantly when separating losses and wins to avoid confounding overlapping FRN and RewP ERPs.”

RESULTS (lines 523-527): “In sum, the large valenced RPE value effect seen when contrasting wins and losses in the FRN/RewP window is composed of the superposition of two separate non-valenced RPE magnitude effects: early frontal theta activity drives negative FRN amplitudes on negative outcomes, and prolonged, more posterior delta activity increases positive RewP amplitudes.”

DISCUSSION (lines 641-651): “Analyses using the standard approach of combining wins and losses implied early frontal theta activity in the FRN epoch represented valenced, scalar RPE value, and subsequent posterior delta band activity at the P3 peak indexed non-valenced RPE magnitude, seemingly supporting a combination of classical RL and independent coding theories^{35,43}. However, repeating the regression analyses for wins and losses separately revealed the early valenced RPE value effect was an artifact of overlap between two distinct non-valenced RPE magnitude effects, providing support for both the salience account^{6,28} and a revised version of the RL theory^{46,47}. Specifically, our data confirm recent studies arguing negative RPEs elicit frontal negativity and theta power consistent with the FRN^{62,63,74,82,83}, while positive RPEs elicit a slower, ramping positivity in delta frequencies consistent with the RewP^{46,50,51,53,61,63,69}.”

DISCUSSION (lines 742-758): “We employed mean window, peak-to-peak, and high-resolution regression analyses across wins and losses aiming to replicate commonly used win-loss difference waves^{38,46,60}. Our results support a combination of the early, classical RL and independent coding theories, mainly that the FRN window represents a scalar, quantitative RPE instead of binary valence and the P3 represents non-valenced RPE magnitude. The overlapping epochs of significance for valenced RPE value and non-

valenced RPE magnitude and probability predictors may also explain why some studies find no difference of valence and instead support the salience account, particularly since mean window and peak-to-peak metrics vulnerable to P3 confounds introduce significant non-valenced effects in our data. However, these approaches failed to capture the true nature of these components.

As suggested by the original authors of the RL theory, interpretations of any analysis that fails to separate wins and losses are flawed because they confound the FRN and RewP contributions on negative and positive outcomes, respectively^{47,55,63}. Instead, temporal and spatial overlap of two distinct quantitative but non-valenced RPE magnitude effects in frontal theta on losses and posterior delta on wins creates the appearance of a valenced RPE value effect.”

Thorough analysis of traditional methods provides an explanation for the conflicting evidence in the literature:

RESULTS (lines 274-279): “The temporal coincidence of significant model coefficients for RPE value and magnitude in the epoch between FRN and P3 peaks suggests that previous findings supporting the salience theory could be explained by component overlap confounds, particularly since FRN and RewP amplitude is commonly quantified as the mean amplitude from approximately 228-344 ms {Sambrook.2015}, an epoch that encompasses both FRN and P3 activity.”

DISCUSSION (lines 758-761): “Note that symmetric deflections in opposite directions caused by the FRN and RewP could potentially cancel out the quantitative tracking of RPE magnitude and explain why some authors observed only binary valence in this time window^{36,43-45}.”

DISCUSSION (lines 747-751): “The overlapping epochs of significance for valenced RPE value and non-valenced RPE magnitude and probability predictors may also explain why some studies find no difference of valence and instead support the salience account, particularly since mean window and peak-to-peak metrics vulnerable to P3 confounds introduce significant non-valenced effects in our data.”

DISCUSSION (lines 769-778): “Importantly, this argument can also explain evidence supporting the salience theory based on paradigm differences in which conditions elicit non-valenced control PEs that trigger MPFC theta responses^{6,28}. ... These results suggest the FRN is not specific to negative RPEs as proposed by the original RL theory but is instead better described by a version of the salience theory in line with action-outcome PEs from the PRO model⁶.”

DISCUSSION (lines 809-813): “These observations also imply the theta response on negative trials may mask the RewP in commonly used difference wave contrasts. Notably, when the RewP is defined as the peak of the win-loss difference wave, it is maximal on the upslope of the P3 and at Cz⁵⁴, which are the precise points of maximal spatial and temporal overlap between FRN and P3 components.”

Identification of the late frontal probability effect and confirmation of this novel finding across two cohorts and in both wins and losses demonstrates the importance of modeling latent cognitive variables and using analysis methods that are unbiased by individual peaks in the ERP waveform.

DISCUSSION (lines 698-702): “...mean window and peak-to-peak measurements are usually aligned to observable peaks, a strategy that would have failed to detect the late probability effect. In sum, these results highlight the benefit of unbiased, high-resolution, multi-dimensional regression analyses for separating and interpreting overlapping components.”

DISCUSSION (lines 818-822): “Our experimental design and modeling approach dissociated outcome probability as a second source of non-valenced salience which has recently been shown to modulate DA coding of RPEs in monkeys⁸⁷. Our multiple regression analyses revealed this predictor captured a novel late frontal positivity in delta frequencies with no observable peak in grand-averaged ERPs and no direct correlation with any Oddball ERPs.”

Collectively, the comprehensive nature of these results illustrate the methods required to dissect interwoven EEG components and relate them to cognitive control and reward learning systems. We hope that our manuscript clearly articulates these methodological and theoretical arguments in such a way that resolves ambiguity regarding previous conflicting results and allows the field to move past debates over early theories and forward in testing more modern ideas.

The second rather notable shortcoming is that there is a substantial amount of literature that uses the single-trial EEG approach employed by the current authors to investigate surprise and/or reinforcement learning. Much of that literature is not discussed (or cited). Some examples of papers that investigated the relationship between P300 waveforms and surprise are Seer et al. (Brain & Cognition, 2016), Mars et al. (J Neuro 2008), Wessel et al. (PLoS Comp Biology 2019). In the domain of the FRN, both Becker et al. (J Neuro 2014) and Meadows et al. (Biological Psychology 2016) have used single-trial analyses, and should likely be cited. So should Talmi et al. (J Neuro 2013), as a major example of a study that doesn't use single-trial EEG, but whose results cast some doubt upon the proposed interpretation of the FRN (and that of Yeung and Sanfey). While perhaps not all of this literature needs to be included, a more comprehensive discussion that couches the current findings into the existing literature (beyond the basics of RL theory) should be undertaken.

We thank the Reviewer for bringing these studies to our attention and have incorporated them into a larger discussion of the methods used in reward processing EEG and their impact on interpretations and theoretical claims.

Minor points

- Why was the behavioral survey only given to a subset of subjects?

We added the post-experiment survey towards the end of collecting the initial cohort and just prior to implementing the Oddball task, so this data is available for 9/15 of the initial cohort and all 17/17 of the replication cohort.

- What criteria were used to evaluate sufficient data quality and behavioral performance to warrant inclusion/exclusion?

We have added the following text to the methods section to clarify the data quality control procedures used to exclude participants and trials.

METHODS for Target Time EEG data (lines 970-975): “Target Time EEG participants were excluded because of technical recording errors (n = 4 datasets with missing EEG or metadata necessary for analysis), excessively noisy data (n = 2 datasets with >3 standard deviation outliers in number of epochs or time points rejected based on visual identification of large, global artifacts), or poor behavioral performance (n = 3 datasets where RT outlier exclusion criteria resulted in <20 trials in any condition), leaving 32 participants for analysis.”

METHODS for subjective ratings behavioral data (lines 965-967): “Participants with prior knowledge of outcome probabilities in the task (n = 2) were excluded after preliminary analyses revealed qualitatively different (less variable) ratings.”

Note that the two excluded participants were lab members that knew details of the experimental design, resulting in qualitatively different rating behavior that was tightly clustered around the true outcome probabilities.

METHODS (lines 1042-1055): “Final quality checks rejected trials for behavioral outliers (RTs missing, < 0.6 s, or > 1.4 s) or EEG artifacts including muscle activity, large voltage shifts, and amplifier saturation identified via visual inspection and using the Fieldtrip function *ft_reject_visual*, resulting in trial counts ranging from 448-524 (mean ± SD: 498.4 ± 20.1). In the remote behavioral subjective rating task, outliers were rejected for the same interval timing response RT criteria and for slow subjective ratings with RTs greater than three standard deviations from the mean, resulting in 382-450 trials (mean ± SD: 437.1 ± 17.6) and 125-150 ratings per participant (mean ± SD: 143.5 ± 6.2). Oddball EEG preprocessing was identical except trials were initially segmented from -0.2 to 1.3 s, and RTs were rejected as outliers if less than 0.1 s or greater than 1.3 s, resulting in trial counts ranging from 334-389 (mean ± SD: 379.5 ± 10.9)”

- The Morlet wavelet for the time-frequency analysis should be described in more detail (cf., MX Cohen, NeuroImage 2019). “

We thank the reviewer for this useful suggestion and have added the full-width at half-maximum (FWHM) of the Morlet wavelets used for time-frequency decomposition in our analyses.

METHODS (lines 1107-1111): “Spectral decompositions were estimated at each time point by convolving the signal with a set of complex Morlet wavelets, defined as complex sine waves tapered by a Gaussian. The frequencies of the wavelets ranged from 1-12 Hz in 1Hz linear steps. The full-width at half-maximum (FWHM) ranged from 1.184-0.096 s with increasing wavelet peak frequency, which corresponds to 3 cycles per frequency.”

Reviewer #2:

In this manuscript, the authors aim to correlate different types of information content with scalp-recorded EEG signals specific to reward and punishment outcomes. The intro proceeds as an interesting and well-informed tour through the field of event-related EEG and different types of prediction errors. However, some critical parts of the theoretical background are outdated and a lot of very important literature is missing. I focus my review on these critical interpretative features and don't focus much on the methods themselves, which are of secondary importance in regards to the conclusions that could be drawn from this task and approach.

Thank you Professor Cavanagh for your careful and thorough critique of our manuscript. Your insights into the nature of reward processing EEG components and the methodological and theoretical debates surrounding their interpretation were invaluable in reframing and improving our manuscript. These comments inspired important new analyses separating wins and losses and shifting of the narrative framework so that readers will come away better informed of the challenges, potential solutions, and remaining uncertainty in the field.

Major Points

- 3) In general, the account of “RL theory” of the FRN here is outdated. This is critical, as the setup (lines 73-75) and conclusion that the original DA hypothesis needs revision (lines 346-348) are not relevant logical foundations. This theoretical revision was long ago suggested by Holroyd himself (PubMed ID: 18513364), where he suggested that the FRN difference wave was due to a generic novelty N2 and a reward-specific component, now called the Reward Positivity. This is of course complicated by the fact that some studies refer to the difference wave as the FRN or as the RewP, whereas other studies use these terms to describe the condition-specific ERP component (FRN=N2, Reward evokes

a RewP alone). Given this messy state of affairs, the authors should clarify their terminologies.

We thank the reviewer for raising these important concerns, as well as for emphasizing the conflicting terminology and definitions in the field. We have expanded our discussion of this theoretical debate to include both classical RL-ERN, salience, and independent coding theories and modern interpretations based on the distinction between the FRN and RewP.

INTRODUCTION (lines 80-101): “An early, influential RL theory (originally called the RL-ERN theory) proposed the FRN represents valenced, quantitative RPE value driven by midbrain DA projections to MPFC³⁵. This hypothesis predicts FRN sensitivity to all the feedback properties that determine RPEs: outcome valence, magnitude, and probability³⁶. However, two recent meta-analyses found mixed evidence for magnitude and probability effects^{37,38}. Reports of larger FRNs to unexpected positive outcomes^{29,39-42} led to an alternative account called the salience theory, which proposes the FRN represents the degree of surprise of an outcome regardless of valence, similar to non-valenced action-outcome PEs driving cognitive control in the PRO model.

A third prominent proposal called the independent coding hypothesis posits the FRN represents binary reward valence instead of scalar RPE value, while the subsequent P3 tracks non-valenced RPE magnitude⁴³⁻⁴⁵. This interpretation is complicated by more recent observations of a P3-like positivity called the Reward Positivity (RewP) that tracks RPE magnitude specifically on positive outcomes (see⁴⁶ for review)⁴⁷⁻⁵² that overlaps in time, space, and delta frequencies with other non-valenced P3 components^{53,54}. Importantly, this suggests losses and wins generate distinct FRN and RewP ERPs with opposite polarities that interact to some degree depending on their overlap in time and space^{48,55}. As a result, it remains unclear after decades of research whether FRN and/or RewP ERPs are driven by valenced RPEs, non-valenced salience PEs, or one of the valenced/non-valenced input variables contributing to these PEs (e.g., outcome valence, magnitude, or probability).”

We have added clear language in the introduction explaining our use of the terms FRN and RewP, in addition to contextualizing their relationships to the broader families of N2-like and P3-like ERPs:

INTRODUCTION (lines 116-121): “Here, we use the term FRN to refer to the early, feedback-locked frontal N2-like ERP that is most prominent on but not specific to losses and the term RewP to refer to the subsequent feedback-locked P3-like ERP specifically following wins. We return to the definitions of FRN and RewP components and their relationship to other ERPs and difference wave contrasts in the discussion.”

INTRODUCTION (lines 59-75): “Seminal early studies identified a posterior scalp positivity generated ~300 ms after detection of an infrequent stimulus called the P3^{18,19} and a fronto-central negativity elicited ~80 ms after incorrect compared to correct responses called the error-related negativity (ERN)^{20,21}. Subsequent extensive literature has revealed these ERPs to be part of large families of similar components. P3-style ERPs are characterized by slow ramping dynamics peaking from ~300-600 ms and in

delta frequencies (~1-4 Hz)²² with at least two different scalp topographies. The P3b has a posterior maximal topography and is elicited by detected events conveying various forms of salient information leading to working memory and model updates, while the P3a has an earlier latency, fronto-central topography and is generated by attention and orienting to novel, task-irrelevant stimuli²³⁻²⁵ (see²⁶ for review). Note that both the P3a and P3b are not unitary physiological events but rather reflect the summed activity of multiple intracranial sources²⁷. ERNs are related to a family of faster latency (~200-300 ms) N2 negativities over fronto-central sensors generated in part by phase-locking in theta frequency (~4-8 Hz) MPFC activity triggered by unexpected events requiring behavioral adjustment²⁸⁻³⁰ (for reviews, see³¹ for stimulus-locked N2s and³² for response-locked ERNs).”

INTRODUCTION (lines 76-80): “Reward feedback conveys multiple informative variables with varying salience and elicits both N2- and P3-like ERPs that overlap in time and space, leading to longstanding debates over which components track different aspects of feedback. Foundational studies focused on the N2-like feedback-related negativity (FRN) occurring ~200 ms at frontal sites after loss feedback compared to wins^{33,34}.”

We now center our arguments on the important issues of disentangling the FRN on losses and RewP on wins, as well as the consequences of methodological choices that confound these effects such as mean window, peak-to-peak, and difference wave metrics. In the Introduction we now note:

INTRODUCTION (lines 102-116): “An important challenge in resolving this debate is disentangling overlapping ERP components. For example, the FRN is commonly measured by averaging ERP amplitude across time, but the epochs used in mean window analyses cover both classic N2 and P3 windows³⁸. The FRN is also often measured by the peak-to-peak amplitude difference between the N2 and the preceding P2 positivity to account for influences of early P3 ramping. However, individual ERP peaks are variable and may not correspond to unique neural sources⁵⁶⁻⁵⁸, rendering their use as reference measures questionable. For example, the P2 shows confounding effects of surprising positive reward^{54,59}. Difference waves are commonly used to isolate target variables such as valence by subtracting ERPs across conditions matched for confounding variables (e.g., magnitude or probability). Indeed, win-loss difference waves are commonly used as the operational definition of the FRN and RewP^{38,54,60,61}. However, this subtraction logic cannot determine which ERP nor which condition were modulated and thus is not well suited to unraveling the dual multiplicity of ERPs and learning signals, particularly since the FRN and RewP may have distinct neural mechanisms supporting different computational roles^{52,62,63}.”

INTRODUCTION (lines 122-128): “The overlap of ERPs in time-domain analyses has made time-frequency decompositions an important tool for separating the FRN and RewP into theta and delta frequencies, respectively. Several studies have shown theta is sensitive to negative RPEs^{51,53,63,64}, but it has also been reported to track non-valenced probability and magnitude⁶⁵⁻⁶⁷. Likewise, delta activity is linked to both non-valenced

surprise^{24,25,68} and positive RPEs^{46,51,53,63,69}, again highlighting the potential mixture of reward specific RewP and P3a and/or P3b ERPs.”

However, this raises other subtle, but critical issues as well. The punishment-evoked FRN (or N2 if you want to call it that) does indeed correlate with negative RPE (20858518, 31149639, others, and so does theta power), but this does not mean that it only encodes this information quantity. In short, FRN and theta are sensitive but not specific to RPE (17850238, 18513364, 22120491, although this becomes complicated when using a difference contrast). In contrast, the reward-specific RewP (i.e. not the difference wave), seems to encode for positive RPE only (25676913), being both sensitive and specific to this information quantity.

We agree with the reviewer that discovering the true nature of the FRN and RewP (or any ERP) must integrate and reconcile findings across studies using diverse paradigms and analysis methods. We believe one strength of our revised manuscript is presenting multiple measures of these components as well as comparisons to N2 and P3 ERPs in the new Oddball task data. We also added language in the discussion addressing the sensitivity and specificity of the FRN and RewP:

DISCUSSION (lines 762-778): “Cavanagh et al. and others have argued the frontal theta response underlying the FRN is an instance of a general MPFC control mechanism elicited by feedback that requires adaptation^{28,29,67,78}. Time estimation tasks like ours employ an implicit win-stay lose-switch strategy in which control adjustments are needed after losses but not wins. Consequently, the FRN mainly appears following negative outcomes, though we do observe weak FRN-like deflections after positive feedback, potentially because vectoral RT feedback allows adjustments of motor timing even when previous responses were correct (e.g., when the target zone is large on easy trials). Importantly, this argument can also explain evidence supporting the salience theory based on paradigm differences in which conditions elicit non-valenced control PEs that trigger MPFC theta responses^{6,28}. For example, FRN and theta responses have been observed following surprising positive outcomes in probabilistic learning tasks when unexpected rewards may indicate shifting reward contingencies that require modifying decision policies, thereby dissociating control PEs from negative valence^{29,41,42,65,66}. These results suggest the FRN is not specific to negative RPEs as proposed by the original RL theory but is instead better described by a version of the salience theory in line with action-outcome PEs from the PRO model⁶.”

DISCUSSION (lines 779-786): “An outstanding question in the reward processing EEG literature is the nature of the RewP and other P3-like components. Our data substantiate claims that RPE magnitude on positive outcomes modulates a delta frequency P3-like component matching the RewP^{46-48,63,69}. This effect appeared strongest in posterior electrodes like a P3b when analyzing wins and losses together, but when examining only positive outcomes, it had a more central distribution closer to a P3a, matching previous reports^{46,51,63,69}. Also, Oddball ERP comparisons showed activity in this epoch correlated

more with the P3a than with the P3b, despite the analysis being conducted at posterior site Pz.”

DISCUSSION (lines 787-817): “Whether the RewP is specific to positive RPEs is complicated by the fronto-central evoked positivity following the FRN on negative outcomes that also increases with RPE magnitude (see Fig. 2 and Sup. Fig. 9). Previous studies analyzing wins and losses together have interpreted these two positive peaks as the same P3 component representing non-valenced RPE magnitude in both conditions along the lines of the independent coding hypothesis^{43,74,86}. However, our data show stronger theta than delta power representations of RPE magnitude at that time point for negative outcomes (see Fig. 6E), as well as rhythmic fluctuations at theta frequencies in frontal RPE magnitude coefficients (Fig. 6A). These observations fit with an alternative interpretation that this positive peak on negative trials is not a P3 but instead due to phase reversal of the theta rhythm underlying the FRN^{53,64}. This later interpretation would also explain why the RPE magnitude effect is stronger for losses in frontal sensors where theta is maximal but stronger for wins at posterior sensors where delta is maximal. Importantly, if the FRN and the following positivity are in fact generated by the same phase reset in MPFC theta activity—which is a more parsimonious explanation—they should not be interpreted as unique components representing different aspects of PEs⁵⁶⁻⁵⁸. These observations also imply the theta response on negative trials may mask the RewP in commonly used difference wave contrasts. Notably, when the RewP is defined as the peak of the win-loss difference wave, it is maximal on the upslope of the P3 and at Cz⁵⁴, which are the precise points of maximal spatial and temporal overlap between FRN and P3 components. Taken together, these data suggest the RPE magnitude effect in our data is not a single non-valenced P3 as posited by the independent coding hypothesis but is due to spatiotemporal overlap between the P3-like RewP specific to positive RPEs and the positivity generated by phase reversal of FRN-linked theta tracking negative RPEs^{53,54,62}.”

In sum, the FRN/N2 (to punishments) and RewP (to rewards) appear to be different signals from different systems that do different things. Current theories suggest such complexities that are indirectly related to the original RL theory (18513364, 25676913), so how does this latest state of affairs influence the author’s theoretical perspective?

Our revised results and interpretations agree with the reviewer’s position that the FRN and RewP are different signals with distinct underlying mechanisms. In light of these new findings, we have updated our interpretations regarding both classical and modern perspectives on these components:

DISCUSSION (lines 868-886): “Multiple regression analyses across temporal, spatial, and frequency dimensions of the data and correlations with canonical N2 and P3 ERPs from the Oddball task elucidate a succession of overlapping components, each corresponding to distinct PEs. We demonstrate the pitfalls of using standard mean window, peak-to-peak, and win-loss difference wave techniques that confound the early,

frontal, theta frequency FRN tracking non-valenced RPE magnitude on negative trials and the concurrent ramping of more posterior delta frequency RewP responses driven by RPE magnitude on positive outcomes. Separating positive and negative outcomes and distinguishing temporal, spatial, and frequency dimensions confirmed an updated version of the salience account of the FRN for negative RPEs that concurs with the PRO model, provided evidence that positive RPEs elicit a P3-like RewP, and identified a novel late frontal P3 tracking low probability outcomes. In summary, we used traditional analyses contrasting wins and losses to reproduce classical evidence of valenced RPE value effects in the FRN window and non-valence RPE magnitude effects in the P3 window that formed the foundations of early RL and independent coding theories. However, follow up analyses separating wins and losses revealed those interpretations were confounded by overlap of three distinct non-valenced salience components. Instead, our data corroborate and extend modern accounts of the FRN as an instance of control PEs generated by MPFC theta and of the RewP as a P3-like component tracking positive RPEs.”

2) The authors need to demonstrate how this work is novel. Point-by-point regressions with unsigned PE and RPE can be seen in many prior studies, for a few examples see work by Philiastides (particularly in regard to signed vs. unsigned PEs evolving over time, e.g. 20510376), Fischer (particularly in regard to the topographic displays, e.g. 24050408), Cavanagh (31149639), Collins (29463751), etc.

We thank the reviewer for these references, which have now been integrated into our review of the literature. We believe our finding of a late frontal positivity tracking outcome probability is novel and distinct from previously reported components.

Aside from this novel component, we believe the comprehensive nature of our work provides important contributions to the debates in the field. Though many studies include subsets of similar analyses or claims, it is only by comparing both traditional and high-resolution single-trial regression results with and without separating wins and losses that we can reveal how the different methods arrive at different conclusions regarding the meaning of these overlapping components. Additionally, our new model comparisons indicate the importance of accounting for reward expectations across conditions, and our comparisons to the N2 and P3 ERPs in our new Oddball data highlight the mixture of components in the critical FRN/RewP epoch. This collection of results directly illustrates how traditional methods produce classical effects like reward valence that are artifacts of component overlap. When combined with our new analyses separating the FRN on losses and RewP on wins, we believe these findings reconcile early and modern accounts.

3) One assumption that this paper makes that is distressingly very common in the field is to consider different ERP components as separate entities. On pg. 5 (and re-raised throughout the results), everything that is said about the relationship between the FRN and the P300 could certainly be true. The FRN and P3 might be unique bumps of evoked activity. Alternatively, the

transition from the peak negativity (FRN) up the slope to the next positivity (P300) could be due to overlapping phase-locked patterns (as could N1 to P2, P2 to N2, etc. see work by Klimesch: 17459593, 15749980), meaning they share at least some variance (likely changing over time). We really don't know the ground truth, but this is reason to be careful and evaluate all possible theoretical accounts.

We agree with the reviewer that peaks in trial-averaged ERPs are not guaranteed to be independent statistical quantities, to correspond to distinct cognitive variables, or to be generated by separate neurobiological mechanisms. Given the importance of this topic in a field often confused about which or even how many components are present, we have added careful language to the discussion on this topic:

INTRODUCTION (lines 107-108): “However, individual ERP peaks are variable and may not correspond to unique neural sources⁵⁶⁻⁵⁸, rendering their use as reference measures questionable.”

Please see above quote from DISCUSSION (lines 787-809) for comments regarding the interpretation of whether the positivity following the FRN on negative outcomes is independent of the FRN.

4) The task appears to evoke a tacit win-stay lose-switch strategy, which is common amongst similar EEG tasks, causing the results to look the same across studies. In such tasks, punishments, but not rewards, require behavioral adaptation, invoking the midfrontal action monitoring system (e.g. 24835663). This diminishes the utility of this specific study for theoretical advancement, since the outcomes will likely look like other WS-LS studies. For example the conclusion drawn on lines 91-92: “This sequence of prediction errors confirms the RL theory of the FRN ...” is far too optimistic and in my view it is both theoretically and methodologically incorrect (see point #1 above). At best, the finding reported here provides a single piece of evidence for one specific theory. But it is a piece of evidence that seems to replicate prior WS-LS findings instead of revealing new ones, and this might be due to the fact that many studies are simply testing the same thing -evoking one type of finding (correlation with signed RPE). The fact that the FRN responds to both signed RPE (citations above) and unsigned RPE (17892382, 21295109, 22120491) suggests that the reality is more complex, and different tasks need to be specifically designed to probe different theoretical accounts.

We thank the reviewer for this insight into how our results relate to those observed in other paradigms. As mentioned above, we have added text in the discussion noting the consequences of the win-stay lose-switch strategy in our task:

Please see above quote from DISCUSSION (lines 762-778).

Reviewer #3:

The authors report results from a single experiment and replication aimed at distinguishing two prominent accounts of dACC/mPFC function - one based on salience, and one based on valenced reward prediction error. To answer this, they manipulate reward probability across separate blocks of a behavioral task in which subjects are required to respond within a visually-cued temporal window while recording EEG, against which they regress model-derived signals, including RPE and unvalenced RPE magnitude.

ERP analysis identifies significant valenced RPE effects associated with the Feedback-Related Negativity (FRN) followed by significant RPE Magnitude effects associated with the P300. They further find significant valenced RPE signatures in the theta band for Fz, and significant RPE magnitude in delta for Pz. Based on these analyses, the authors conclude that the valenced RPE account of the FRN is fundamentally correct, but in need of revision to incorporate a larger diversity of reward-signals.

mPFC function remains an ongoing topic of debate, especially w/r/t the role of the region in salience or surprise vs processing value information, and I appreciate the authors' efforts to quantitatively test these competing accounts. However, I have major reservations regarding the computational methods.

We thank the reviewer for their recognition of the importance of comparing surprise and RPE-based accounts of MPFC activity and our work to quantitatively compare these theories. We hope that the revised analyses better delineate the underlying EEG components and further detail how they argue against previous explanations based on outcome features and valence. Please see below for our theoretical and empirical responses to your reservations.

The authors model behavior (accuracy) with a logistic function to derive trial-by-trial Win probabilities. From there, they derive value-based quantities, including EV, RPE, and salience (absolute RPE). Although these values derive from the observed behavior of the subjects, they do not necessarily reflect the subjective beliefs about behavior. Subjects could be very bad at tracking the objective probabilities (e.g., Hajcak et al., 2007), and therefore the values derived from objective behavior would be irrelevant to the FRN.

We thank the reviewer for raising this important issue. One of the core claims in our manuscript is that reward processing EEG signatures should be analyzed and interpreted relative to predictions. We address this issue using model comparisons in the Results, and also in the new section of the Discussion titled “Theoretical implications: Predictive coding”:

RESULTS (lines 300-314): “Figure 2E shows that RL-based models including RPE value capture more variance in EEG amplitude during the FRN window at Fz than outcome value or valence models (see Sup. Fig. 4C-F for outcome-based model coefficients; see Sup. Table 2 for AIC model comparison value averaged within peak model coefficient windows). The model with binary outcome valence performs better

than the outcome-based model with scalar value, and these model results hold for both mean window and peak-to-peak estimates of FRN amplitude (Sup. Fig. 3E and 3F). The only difference between outcome value and outcome valence models is whether neutral trials in easy and hard blocks are treated as outcomes with identical values (zero) or as omissions of expected rewards with opposite valence (1 or -1). Similarly, Figure 2F shows that the RL-based models outperform the outcome value and valence models at Pz throughout the FRN and P3 epochs, which is confirmed by model comparisons using the mean window estimates of P3 amplitude in Supplemental Figure 3G. These results confirm that FRN and P3 ERPs are best viewed through the predictive coding lens of PEs.”

RESULTS (lines 594-598): “Reward expectations are just as critical to RPEs as the outcome, and failure to account for differences in these predictions across paradigms contributes to the disagreements in the reward EEG literature. Previous work has indicated participants’ subjective reward expectations may deviate from objective probabilities established by the experimental design or modeled from behavior ^{39,70}.”

DISCUSSION (lines 705-724): “Our results emphasize the importance of accounting for reward expectations when interpreting reward processing EEG. Quantitative model comparisons showed a model comprised of outcome value, magnitude, and probability—features commonly associated with the FRN and RewP but that do not account for expectations—performed worse than the RL-based model across all ERP features. The outcome-based model was improved when outcome value was replaced with outcome valence to account for different reward expectations on neutral trials in easy and hard conditions, though this outcome valence model was still worse than the RL model based on PEs. The importance of predictive coding was especially apparent on neutral trials with identical outcomes but different feedback valence based on expectations. FRN peak latencies and frontal theta phase were modulated by RPE value, and similarly, FRN peak latency was earlier for neutral feedback with negative than positive RPEs. However, Williams et al. found an opposite FRN latency shift with wins occurring earlier than losses in a large (n=500) gambling dataset using only visual feedback ⁵⁴. The latency of FRN and RewP ERPs in our data are also earlier than reported in recent meta-analyses ³⁸, likely because feedback in our task includes auditory components which generate faster FRN latencies ³³. These discrepancies indicate that although our latency and phase results provide strong evidence of the influence of reward predictions on neutral trials, they may not generalize to paradigms with different feedback properties.

In contrast to these different brain responses to neutral outcomes, post-experiment survey data indicated participants’ explicit interpretations of neutral feedback did not differ across easy and hard conditions. This finding supports assertions that the FRN is generated by a habitual, model-free reward learning circuit that bypasses conscious representations of goal-directed task structure and instead relies on implicit associative learning mechanisms ^{37,71}. However, previous work indicates that failure to account for subjective expectations can obscure reward EEG effects and confound their interpretation ⁷⁰. To address this issue, we collected subjective ratings of win probabilities in an additional behavioral experiment that showed participants’ subjective expectations tracked our difficulty manipulation and their likelihood of winning or losing. We

employed a control RL model incorporating subjective biases representative of those ratings to eliminate concerns that differences between subjective and model-based reward expectations influenced our EEG conclusions.”

We also agree that if participants’ subjective expectations deviate dramatically from the objective reward contingencies in our task, then our results and conclusions would be flawed. However, we believe the transparent nature of feedback statistics in our task render it less vulnerable to these issues than probabilistic gambling or reinforcement learning paradigms in the literature. In particular, participants receive a visual cue at the start of every trial (the target zone) which accurately conveys win probability and is directly determined by performance, and feedback is completely veridical and contains both outcome and RT information needed to adjust future responses. Therefore, we believe participants in our Target Time task have greater insight into reward expectations than probabilistic gambling or learning tasks such as the one used in the Hajcak et al. 2007 study.

Perhaps more importantly, Ichikawa et al. (2010, Int. J. Psychophysiology) directly compared how well RPEs derived from subjective ratings and behavioral modeling using an RL algorithm were able to predict the FRN. They found that the FRN was better predicted by RL models of behavior than subjective ratings (see their Fig. 6), consistent with this being an early, non-conscious model-free learning signal that doesn’t always correlate with behavioral adjustment (Walsh & Anderson, 2012). In addition to these theoretical arguments, we address this concern directly with new empirical data described below.

Some models based on the salience account (such as the PRO model cited by the authors) explicitly address how differences in subjective estimates of probability, deriving from, e.g., relative sensitivity to reward or error feedback, could lead to changes in the magnitude of error feedback in mPFC. If, as in the Hajcak study, subjects in this study overestimated their performance in the HARD condition, it would be possible to observe unvalenced RPEs that correlate strongly with (objective) valenced RPEs. Since the main point of the manuscript is to adjudicate between the salience and RL theories, it is essential that the subjective probability estimates be used to calculate RPEs. I don't see how that is possible with this study, however.

Despite the theoretical arguments provided above, we agree it is still possible that participants have poor insight into their performance in our Target Time task. Although we could not collect new EEG data due to the ongoing COVID-19 pandemic, we addressed this concern by collecting a new remote behavioral dataset (n = 24) that directly measures subjective ratings of reward probability using a slider bar on every third trial after responding but before receiving feedback. Control analyses designed to address the reviewer’s concerns are reported in the new results section “ERP PE sequence results are robust to biases in subjective reward expectations” from lines 626-667, which we summarize below.

Supplemental Figure 10A shows the logistic fit to a representative participant as in Fig. 1D except with their subjective rating data overlaid in green for wins and red for losses. Ratings clearly differentiate between easy and hard conditions.

Indeed, at the group level, ratings correlate strongly with model-based win probabilities across conditions (see Sup. Fig. 10B), indicating participants are sensitive to the main difficulty manipulation in the task. This correlation is also significant in hard conditions, though it is not significant in easy conditions, potentially since less effort and attention is required when accuracy is ~80%.

RESULTS (lines 602--606): “Comparing these subjective ratings to those derived from our behavioral modeling revealed strong correspondence overall ($r = 0.705, p < 10^{-10}$; Sup. Fig. 10B), but this relationship was likely driven by the large difference between easy and hard conditions, as the correlation was significant only within hard conditions ($r = 0.123, p < 10^{-6}$) and not within easy conditions ($r = 0.007, p = 0.79$).”

Furthermore, participants rated their probability of winning significantly higher before correct than incorrect outcomes in easy blocks, hard blocks, and all conditions combined. These results indicate subjective ratings are sensitive to both experimental manipulations of difficulty and behavioral performance and confirm participants had substantial conscious insight into the reward contingencies in the task, indicating subjective probability estimates and similar to our model-based estimates in this task.

RESULTS (lines 606-612): “Nonetheless, participants rated their probability of winning significantly higher before wins than losses overall ($t(3156) = -4.27, p < 10^{-4}$), as well as within only easy ($t(1575) = -3.52, p < 10^{-3}$) and only hard conditions ($t(1579) = -4.29, p < 10^{-4}$). These data suggest that despite variance in subjective ratings, participants are sensitive to both the experimental manipulation of difficulty and their own behavior which combine to form reward expectations in our task.”

Nevertheless, participants did exhibit some degree of subjective bias such that they underestimated their probability of winning in the easy condition and they overestimated their probability of winning in the hard condition, as suggested by the Reviewer (see Sup. Fig. 10C).

RESULTS (lines 616-622): “However, the rating data from our behavioral experiment indicate participants’ subjective expectations were biased relative to our behavioral model such that they overestimated their likelihood of winning in hard conditions and underestimated it in easy conditions (Sup. Fig. 10C). The biases measured in our behavioral experiment are small on average (mean \pm SD in hard: $-8.1 \pm 10.2\%$; mean \pm SD in easy: $6.3 \pm 11.5\%$), but the results of our regression analyses could be affected by subjective biases in our sample of EEG participants.”

Since it was not possible to measure subject specific biases in a new EEG dataset, we used simulations to test whether our EEG results would be affected by biases representative of those measured in the new remote behavioral cohort. Because subjective win probability ratings in both easy and hard conditions are biased towards chance (50%), incorporating both of these terms in our model eliminate differences in reward expectations across conditions such that expected value is equivalent for easy and hard conditions (see black Expected Value predictor below). This effectively recreates the outcome value model that does include any reward expectations (compare model predictions in Sup. Fig. 1B and 1D), and results using these parameters are already reported in Sup. Fig. 4C and 4D. Importantly, model comparisons in Fig. 2 and Sup. Table 2 demonstrate this model does not explain the EEG data as well as our RL model, but regardless, incorporating both easy and hard biases into our model does not confound our results.

RESULTS (lines 622-627): “To address this potential confound, we simulated the effect of adding similar biases to the expected value predictor in our model. Due to the opposite directionality of bias in easy and hard conditions, adding the most extreme bias observed in our behavioral cohort (+/-25% shift in win probability) equalized reward expectations across easy and hard conditions and effectively reproduced the outcome value model tested above (compare model predictions in Sup. Fig. 1C to those in Sup. Fig. 1B).”

To more directly address the reviewer’s concern and previous results (e.g., Hajcak et al., 2007) that participants are overconfident in the hard condition, we also included a model with only bias in the hard condition (see model predictions in Sup. Fig. 1C).

EEG results were similar to RL models without any bias (see Sup. Fig. 4K and 4L).

RESULTS (lines 627-634): “Since the optimistic bias in the hard condition was slightly larger and matches an overoptimistic bias reported in previous studies ⁷⁰, we also simulated a model with +25% shift in win probability for only hard conditions (Sup. Fig. 1D). Repeating our multiple regression analyses at Fz and Pz with the RL model including subjective bias in hard conditions reproduced all of the main RPE value, RPE magnitude, and probability results from Fig. 2 (Sup. Fig. 4K and 4L). Overall, these analyses show that the level of subjective bias in reward expectations in our task do not affect our results and conclusions.”

Finally, as predicted by the Reviewer, correlations between the valenced RPE value and non-valenced RPE magnitude regressors do increase from $r = 0.055$ in the regular RL model to $r = -0.524$ when adding the hard subjective bias to our model, but the variance inflation factors are below 2 (maximal $VIF_{RPE_{magnitude}} = 1.67$), indicating the Reviewer’s concerns should not influence our conclusions.

In summary, although we agree with the reviewer that deviations in subjective reward expectations from expected value based on actual performance is an important issue in the field, we believe that this issue is not a problem in our data and results based on the theoretical and empirical arguments enumerated above.

Beyond this major concern, however, the manuscript is well-organized and clearly stated. If the issue with subjective vs objective probability can be resolved in a revision, the results presented would provide additional perspective to a long-running debate in the cognitive control literature.

We thank the Reviewer for their kind words and support and hope our new subjective rating data and analyses extrapolating from those results address your concern regarding subjective vs. objective probabilities in reward expectations.

Reviewers' comments:

Reviewer #1 (Remarks to the Author):

The authors have done an exhaustive and commendable job addressing my original review, and - in my estimation - the excellent points put forward by the other two reviewers. I have no remaining points of criticism and congratulate the authors on a very informative and comprehensive paper.

Reviewer #2 (Remarks to the Author):

The authors have been very responsive to the Reviewer's concerns.

Very Minor Points:

1) There are some very technical, minute adjustments I suggest for the depiction of the RewP:

Line 94: I think it's incorrect to assert that the RewP overlaps in *time* with other non-valenced P3 components. Space and frequency - yes. But the RewP is reliably earlier than the P3a, which nearly always follows an N2 (and the RewP canonically overlaps the N2). As an aside, the RewP is reliably earlier than that when auditorily cued, as the authors show here (also see PMID: 32450114). The P3a displays some different characteristics to sound, namely it still tends to follow an N2 and still occurs later than the RewP.

Line 126: "Likewise, delta activity is linked to both non-valenced surprise... and positive RPEs ... again highlighting the potential mixture of reward specific RewP and P3a and/or P3b ERPs." Here I again urge caution in describing the RewP and P3s as similar simply due to the fact that they both have delta band spectral reflections.

2) The Results are quite long and some new sections don't seem to add to the manuscript.

In the section: *_FRN peak latency and frontal theta phase shift with RPE valence_*, the findings don't add to the take-home message. This results section is rather complex and confusingly reported. In addition, these outcomes are possibly biased due to natural artifactual contamination in the trials: a downslope of a positive-expectation process on neutral trials (i.e. a RewP to neutral cues occurs because they are better than bad outcomes) can overlap a canonical P2-N2-P3 process and can create combined features that look like an N2, but are not likely canonical as the oddball data would show. In short, I'd urge caution in this unclear reporting of a complex analysis on tenuous data that doesn't appear to add anything critical to the overall conclusions.

I think the section: *_ERP PE sequence results are robust to biases in subjective reward expectations_* is a nice addition, but it could probably be reported in the main text with the last 2 sentences, where the rest could be added to the supplement.

Reviewer #3 (Remarks to the Author):

I thank the authors for their extensive response to my major concern, which includes new behavioral data, model results, and EEG analyses. In these new analyses, the authors find that subjective ratings of win probability can be quite variable - the author's report that the 'most extreme' bias is a +/- 25% win probability shift (overestimating win probability in hard conditions, underestimating in easy

conditions). It appears that this 'extreme bias' is averaged over the trials for a single subject; the difference between subjective estimates and model-estimated win probability for individual trials seems like it can be quite a bit larger (Fig S10B).

The authors include the extreme bias to develop additional model predictions - in one case, the extreme bias is applied to both hard and easy conditions (essentially recreating the outcome model) and in another, the bias is applied only to the hard condition, producing a correlation between valenced RPEs and salience PEs. They then repeat the analyses of their EEG data using predictions from the model with subjective bias in the hard condition, and are able to recover effects observed from their analysis using the version without subjective bias.

My understanding is that in order to derive subjectively biased predictions, the bias was applied across the board - regardless of whether the trial outcome was win or lose, the modified 'subjective' win estimate was adjusted by 25%. As the authors note in their response, beyond an optimistic bias on hard trials (and a pessimistic bias on easy trials), subjective win probability appears to reflect subjects' estimate of their own behavioral performance: subjective win probability is higher before wins than losses in both hard and easy condition.

For me this raises the question of whether including a uniform subjective bias by condition is sufficient to control for possible correlations between valenced and unvalenced RPEs. For example, Fig S10A shows a subject with positive/negative biases in hard/easy conditions, respectively. However, (at least for this subject) the negative bias in the easy condition appears to be at least partially driven by lower subjective win estimates preceding loss trials. Given the FRN's sensitivity to expectations, trials in which a subject accurately identifies their own poor performance (and thus may have a higher expectation of negative feedback) may elicit weaker error activity than trials in which negative feedback is more surprising.

I don't think this concern is sufficient to preclude publication; the authors have done a nice job in addressing my original concern to the extent that it is possible. I do, however, think it is appropriate to note the issue in the discussion as a possible deficit.

Minor:

In their response, the authors state that Ichikawa et al (2010) find that the FRN is better explained by RL models than subjective ratings (fig. 6); this doesn't seem like an accurate characterization. Ichikawa et al find moderate/high correlations between both RL RPEs and subjective RPEs with the FRN (0.4 & 0.47, respectively), but no overall significant difference between the correlations. This doesn't really have an impact on the manuscript, just making a note.

Response to Reviewers:

We thank the reviewers and editorial office for their additional comments. Please see below for a **point-by-point response in bold** to each *reviewer's comments, which are underlined in italics*. Please note that **new text added to the manuscript is highlighted in blue**.

Reviewer #1:

The authors have done an exhaustive and commendable job addressing my original review, and - in my estimation - the excellent points put forward by the other two reviewers. I have no remaining points of criticism and congratulate the authors on a very informative and comprehensive paper.

We thank the reviewer for their kind words and appreciation of the paper.

Reviewer #2:

The authors have been very responsive to the Reviewer's concerns.

We thank the reviewer for their acknowledgement of our efforts, their continued attention to detail, and their overall contributions to improving the manuscript.

Very Minor Points:

1) There are some very technical, minute adjustments I suggest for the depiction of the RewP:

*Line 94: I think it's incorrect to assert that the RewP overlaps in *time* with other non-valenced P3 components. Space and frequency - yes. But the RewP is reliably earlier than the P3a, which nearly always follows an N2 (and the RewP canonically overlaps the N2). As an aside, the RewP is reliably earlier than that when auditorily cued, as the authors show here (also see PMID: 32450114). The P3a displays some different characteristics to sound, namely it still tends to follow an N2 and still occurs later than the RewP.*

We thank the reviewer for highlighting these distinctions. We have removed the language about temporal overlap, resulting in the following statement:

INTRODUCTION (lines 92-95): “This interpretation is complicated by more recent observations of a P3-like positivity called the Reward Positivity (RewP) that tracks RPE magnitude specifically on positive outcomes (see ⁴⁶ for review) ⁴⁷⁻⁵² that overlaps in space and delta frequencies with other non-valenced P3 components ^{53,54}.”

Line 126: “Likewise, delta activity is linked to both non-valenced surprise... and positive RPEs ... again highlighting the potential mixture of reward specific RewP and P3a and/or P3b ERPs.” Here I again urge caution in describing the RewP and P3s as similar simply due to the fact that they both have delta band spectral reflections.

We agree about the importance of clarity when describing the similarities and differences between these components and have revised this sentence to emphasize the general issue of separating components using a single metric/dimension rather than the shared frequency signature of the RewP and P3s:

INTRODUCTION (lines 127-130): “These mixed results highlight how individual measures of neural activity may be insufficient to distinguish between ERP components such as the RewP, P3a, and P3b that overlap in one or more dimensions (e.g., frequency) but correspond to distinct cognitive variables.”

2) The Results are quite long and some new sections don't seem to add to the manuscript.

We appreciate the reviewer's advice on clarifying the take home message for readers.

In the section: *FRN peak latency and frontal theta phase shift with RPE valence*, the findings don't add to the take-home message. This results section is rather complex and confusingly reported. In addition, these outcomes are possibly biased due to natural artifactual contamination in the trials: a downslope of a positive-expectation process on neutral trials (i.e. a RewP to neutral cues occurs because they are better than bad outcomes) can overlap a canonical P2-N2-P3 process and can create combined features that look like an N2, but are not likely canonical as the oddball data would show. In short, I'd urge caution in this unclear reporting of a complex analysis on tenuous data that doesn't appear to add anything critical to the overall conclusions.

We aimed to report the difference in ERPs to neutral feedback driven solely by reward expectations and RPE valence to convey the importance of accounting for predictions when analyzing reward processing EEG signals. However, we agree with the reviewer this section is unnecessarily complex to achieve that goal. To simplify this point, we have removed the phase analyses and results altogether since they are redundant and capture the same effect as the latency shift. We have also moved the latency shift results to Supplementary Note 1, leaving a single sentence in the results to emphasize the impact of reward expectations on reward ERPs:

RESULTS (lines 558-561): “For example, comparing ERPs following easy and hard neutral trials with identical feedback but opposite reward expectations showed FRN latency shifts according to RPE valence that matched those observed in losses and wins, respectively (see Supplementary Note 1).”

See **SUPPLEMENTARY METHODS, SUPPLEMENTARY NOTE 1 (“FRN peak latency shifts with RPE valence”)**, and **SUPPLEMENTARY FIGURE 9** for reporting of the analyses and results underlying this statement.

We have also rephrased the interpretation of these results in the Discussion to address the reviewer's point about the plurality of potential causes of this effect in the context of multiple overlapping components:

DISCUSSION (lines 657-661): “These discrepancies and the presence of multiple overlapping components indicate that although our latency results provide evidence of the influence of reward predictions on neutral trials, they should be interpreted with

caution and may not generalize to paradigms with different feedback stimuli or task demands^{58,60,85}.”

I think the section: ERP PE sequence results are robust to biases in subjective reward expectations is a nice addition, but it could probably be reported in the main text with the last 2 sentences, where the rest could be added to the supplement.

We thank the reviewer for their appreciation of these new data, analyses, and results, and we agree that the purpose of this section can be achieved with shorter text while the details are moved to the supplement. Accordingly, we have combined these results with the above sentence on FRN latency and reward expectations into the following paragraph:

RESULTS (lines 556-571): “Reward expectations are as critical to RPEs as the outcome, and failure to account for differences in these predictions across paradigms contributes to the disagreements in the reward EEG literature. For example, comparing ERPs following easy and hard neutral trials with identical feedback but opposite reward expectations showed FRN latency shifts according to RPE valence that matched those observed in losses and wins, respectively (see Supplementary Note 1). Furthermore, previous work has indicated participants’ subjective reward expectations may deviate from objective probabilities established by the experimental design or modeled from behavior^{39,70}. To assess whether this issue impacted our results, we collected an additional behavioral dataset to measure subjective ratings of reward expectations before feedback (see Supplementary Methods). Ratings tracked difficulty (easy/hard) and trial outcomes (win/loss) and revealed subjective biases such that participants underestimated their probability of winning in easy conditions and overestimated it in hard conditions (Supplementary Note 2 and Sup. Fig. 10). The EEG results in Fig. 2 were reproduced when incorporating similar biases into our RL model, providing evidence that our conclusions are robust to differences between subjective and model-based reward expectations.”

See **SUPPLEMENTARY NOTE 2** (“ERP PE sequence results are robust to biases in subjective reward expectations”) and **SUPPLEMENTARY FIGURE 10** for reporting of the analyses and results underlying this paragraph.

Reviewer #3:

I thank the authors for their extensive response to my major concern, which includes new behavioral data, model results, and EEG analyses.

We thank the reviewer for their acknowledgement of our efforts to measure these effects in a new behavioral dataset and test their impact on our findings with new modeling analyses.

In these new analyses, the authors find that subjective ratings of win probability can be quite variable - the author's report that the 'most extreme' bias is a +/- 25% win probability shift (overestimating win probability in hard conditions, underestimating in easy conditions). It appears that this 'extreme bias' is

averaged over the trials for a single subject; the difference between subjective estimates and model-estimated win probability for individual trials seems like it can be quite a bit larger (Fig S10B).

The authors include the extreme bias to develop additional model predictions - in one case, the extreme bias is applied to both hard and easy conditions (essentially recreating the outcome model) and in another, the bias is applied only to the hard condition, producing a correlation between valenced RPEs and salience PEs. They then repeat the analyses of their EEG data using predictions from the model with subjective bias in the hard condition, and are able to recover effects observed from their analysis using the version without subjective bias.

My understanding is that in order to derive subjectively biased predictions, the bias was applied across the board - regardless of whether the trial outcome was win or lose, the modified 'subjective' win estimate was adjusted by 25%. As the authors note in their response, beyond an optimistic bias on hard trials (and a pessimistic bias on easy trials), subjective win probability appears to reflect subjects' estimate of their own behavioral performance: subjective win probability is higher before wins than losses in both hard and easy condition.

For me this raises the question of whether including a uniform subjective bias by condition is sufficient to control for possible correlations between valenced and unvalenced RPEs. For example, Fig S10A shows a subject with positive/negative biases in hard/easy conditions, respectively. However, (at least for this subject) the negative bias in the easy condition appears to be at least partially driven by lower subjective win estimates preceding loss trials. Given the FRN's sensitivity to expectations, trials in which a subject accurately identifies their own poor performance (and thus may have a higher expectation of negative feedback) may elicit weaker error activity than trials in which negative feedback is more surprising.

We agree that the optimal way to measure the effect of subjective expectations on reward processing ERPs would be to acquire single-trial reward predictions with concurrent EEG. Unfortunately, collecting such data was not possible during the COVID-19 pandemic, forcing us to resort to remote behavioral data collection online and limiting our control analyses to block-level estimates of bias per condition.

I don't think this concern is sufficient to preclude publication; the authors have done a nice job in addressing my original concern to the extent that it is possible. I do, however, think it is appropriate to note the issue in the discussion as a possible deficit.

We again thank the reviewer for their recognition of our efforts to address their concern and have added two sentences in the discussion articulating this limitation and calling for future studies to address this important issue.

DISCUSSION (lines 674-677): “However, these behavioral data and modeling analyses do not account for differences in subjective biases across participants or predictions on individual trials (e.g., accurately identifying errors on easy trials before feedback). Future studies may define how these effects modulate EEG signatures of PEs.”

Minor:

In their response, the authors state that Ichikawa et al (2010) find that the FRN is better explained by RL models than subjective ratings (fig. 6); this doesn't seem like an accurate characterization. Ichikawa et al find moderate/high correlations between both RL RPEs and subjective RPEs with the FRN (0.4 & 0.47, respectively), but no overall significant difference between the correlations. This doesn't really have an impact on the manuscript, just making a note.

We thank the reviewer for their careful reading and interpretation of our response and for bringing this point to our attention.

REVIEWERS' COMMENTS:

Reviewer #2 (Remarks to the Author):

I have no further concerns. The manuscript should be published as-is.

Reviewer #3 (Remarks to the Author):

The authors have addressed my last remaining concerns and I believe the manuscript is ready for publication.